# The Remarkable Robustness of LLMs: Stages of Inference?

**Vedang Lad**[*]
MIT
Stanford University
vedanglad@stanford.edu

**Jin Hwa Lee**
University College London
jin.lee.22@ucl.ac.uk

**Wes Gurnee**
MIT
wesg@mit.edu

**Max Tegmark**
MIT
tegmark@mit.edu

## Abstract

We investigate the robustness of Large Language Models (LLMs) to structural interventions by deleting and swapping adjacent layers during inference. Surprisingly, models retain 72–95% of their original top-1 prediction accuracy without any fine-tuning. We find that performance degradation is not uniform across layers: interventions to the early and final layers cause the most degradation, while the model is remarkably robust to dropping middle layers. This pattern of localized sensitivity motivates our hypothesis of four stages of inference, observed across diverse model families and sizes: (1) detokenization, where local context is integrated to lift raw token embeddings into higher-level representations; (2) feature engineering, where task- and entity-specific features are iteratively refined; (3) prediction ensembling, where hidden states are aggregated into plausible next-token predictions; and (4) residual calibration, where irrelevant features are suppressed to finalize the top-1 output distribution. Synthesizing behavioral and mechanistic evidence, we provide a hypothesis for interpreting depth-dependent computations in LLMs.

## 1 Introduction

Recent advancements in Large Language Models (LLMs) have exhibited remarkable reasoning capabilities, often attributed to increased scale [1]. Understanding these capabilities and mitigating associated risks [2–4] have motivated extensive research into their underlying mechanisms.

A *bottom-up* approach to interpretability, known as mechanistic interpretability, has explored the iterative inference hypothesis [5, 6], which posits that each transformer layer incrementally updates a token's hidden state toward minimizing loss by progressively shaping the next-token distribution [7]. This is supported by self-repair [6], where later layers correct or mitigate errors introduced by earlier layers, and redundancy [8, 9], where multiple layers perform similar or overlapping computations to refine predictions.

It remains unclear how this iterative view of inference fits with the "circuit" hypothesis, which argues for clearly delineated, specialized roles for certain model components. This is supported by induction heads [10], successor heads [11], copy suppression mechanisms [12], and knowledge neurons [13], among other "universal" neurons [14, 15]. Whereas iterative inference suggests gradual refinement through overlapping computations, the strong circuit hypothesis implies distinct, modular computational pathways.

---

[*]Corresponding author. Work started at MIT, completed at Stanford University

39th Conference on Neural Information Processing Systems (NeurIPS 2025).

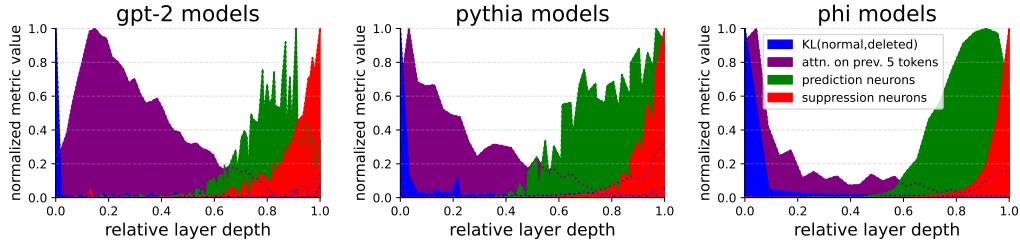

Figure 1: Statistical signatures of stages of inference averaged across three model families. (Blue) KL between the normal model and layer $\ell$ zero-ablated. (Purple) Total attention paid to the previous five tokens in a sequence. (Green) The number of "prediction" neurons (Red) The number of "suppression" neurons [20, 15, 14].

Table 1: Our Hypothesis: Stages of Inference

| Stage | Name | Function | Observable signatures |
|-------|------|----------|----------------------|
| 1 | Detokenization | Integrate local context to transform raw token representations into coherent entities | Catastrophic sensitivity to deletion and swapping and attention-heavy computation. |
| 2 | Feature Engineering | Iteratively build feature representation depending on token context | Little progress made towards next token prediction, but significant increase in probing accuracy and patching importance. |
| 3 | Prediction Ensembling | Convert previously constructed semantic features into plausible next token predictions using an iterative ensemble of model components | Prediction neurons appear and output distribution begins to narrow. |
| 4 | Residual Calibration | Eliminate obsolete features and form the next token distribution from internal representation | Suppression neurons appear and output distribution shapes for top-1 prediction with a growing MLP-output norm |

Naturally, layer-wise phenomena in LLMs are also documented outside formal interpretability research and provide more evidence to existing interpretability findings. For example, while knowledge storage within mid-layer MLP neurons has been demonstrated [16], other non-interpretability work has found that fine-tuning predominantly affected the weights in the middle layers [17]. Quantization studies identified improved benchmark performance by retaining only low-rank MLP components from the middle to later layers [18]. Other works have noted a transition in activation sparsity from sparse to dense around mid-model depth [19, 15]. These behavioral findings, when integrated with mechanistic insights, suggest a layered computation structure not yet fully characterized.

To explore this structure, we perform layer-wise interventions—deleting individual layers or swapping adjacent ones (Figure 13)—to characterize their localized effects. Building on these insights, we analyze depth-wise roles and synthesize our findings with prior interpretability work to propose a four-phase hypothesis that attempts to bridge the top-down and bottom-up views of computation in decoder-only LLMs.

Concretely, we hypothesize four depth-dependent stages: **(1) detokenization**, **(2) feature engineering**, **(3) prediction ensembling**, and **(4) residual calibration**. In brief, early layers integrate local context to form coherent entities; middle layers iteratively construct features; later layers convert these features into next-token predictions via an ensemble of neurons. Figure 1 and Table 1 summarize these stages and their associated empirical signatures. We synthesize these findings with prior interpretability work [21] to suggest a depth-aligned computational structure in LLMs.

## 2  Related Work

**Mechanistic Interpretability**    Mechanistic interpretability often employs circuit analysis to uncover model components relevant to specific computations. In computer vision, universal mechanisms such as frequency detectors and curve-circuits have been identified [22–24], with features become progressively more complex through the layers of CNNs. These principles were later extended to modern transformers [25, 26], where similar circuit-based analyses revealed phenomena such as circuit reuse [27], variable-finding mechanisms [28], self-repair [6, 29], function vectors [30, 31], and long-context retrieval [32].

**Iterative Inference and Depth-Dependent Computations**    The iterative inference hypothesis, first explored in ResNets [33, 34], posits that each layer incrementally updates token representations. This idea has gained traction in transformers, particularly through logit lens analyses [35, 5], which visualize the model's evolving prediction distributions layer by layer. Some studies further suggest discrete inference phases [36], with certain computations localized to specific depths—such as truth-processing [37] or multilingual translation [38]. These findings are complemented by layer permutation studies showing that performance improves when self-attention layers precede feedforward layers [39].

**BERTology**    Prior work on ablations and layer-wise analysis has primarily focused on BERT [40]. These studies reveal substantial redundancy: even with aggressive neuron and layer pruning, models retain most of their performance [41–45]. More recent investigations corroborate this, showing that a significant portion of attention heads and feedforward components can be removed with minimal accuracy loss [9, 8].

## 3  Experimental Protocol

Table 2: Comparison of Language Model Architectures

| Model Series | Size | Layers | Model Series | Size | Layers |
|---|---|---|---|---|---|
| Pythia | 410M | 24 | Microsoft Phi | Phi-1 (1.3B) | 24 |
| | 1.4B | 24 | | Phi-1.5 (1.3B) | 24 |
| | 2.8B | 32 | | Phi-2 (2.7B) | 32 |
| | 6.9B | 32 | Llama 3.2 | 1B | 16 |
| | 12B | 36 | | 3B | 28 |
| GPT-2 | Small (124M) | 12 | Qwen 2.5 | 0.5B | 24 |
| | Medium (355M) | 24 | | 1.5B | 28 |
| | Large (774M) | 36 | | 3B | 36 |
| | XL (1.5B) | 48 | | 14B | 48 |

**Models**    To investigate the stages of inference in language models, we examine the Pythia [46], GPT-2 [47], Qwen 2.5 [48], LLaMA 3.2 [49], and Microsoft Phi [50, 51] model families, which range from 124M to 14B parameters (see Table 2). All families use decoder-only transformers but differ in their execution of attention and MLP components. Specifically, Pythia models execute attention and MLP layers in parallel. In contrast, GPT-2, Phi, and Llama models apply attention followed by an MLP sequentially. We preprocess weights identically across all models, folding in the layer norm, centering the unembedding weights, and centering the writing weights as described in Appendix B. Despite these architectural differences, most phenomena remain consistent across models, though we discuss drawbacks in Limitations 6.

**Data**    Besides data agnostic experiments, we evaluate all five model families on a corpus of one million tokens from random sequences of the Pile dataset [52], unless otherwise noted in the experiment.

**Layer Swap Data Collection**    To study the robustness and role of different model components at different depths, we employ a swapping intervention where we switch the execution order of a

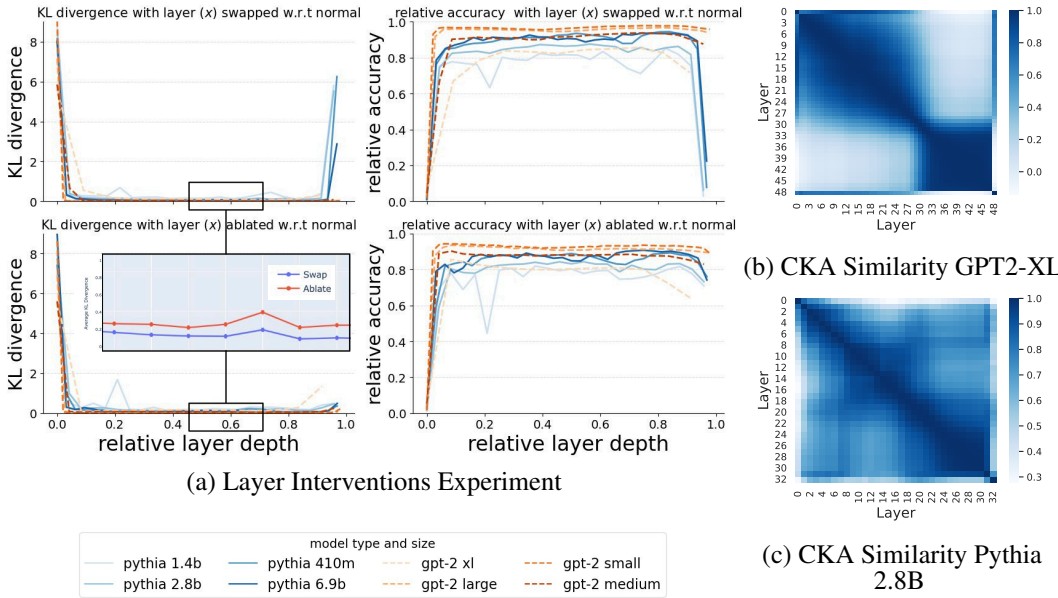

Figure 5: (a) Effect of layer swap (top) and layer drop (bottom) interventions on model behavior. (left) KL divergence between the intervened and original models. (right) Consistency of top-1 predictions. (b)(c) Representational similarity across layers measured using CKA, showing block-like structure in GPT-2 XL (b) and Pythia 2.8B (c). Similar trends are observed across other model families and sizes (see Appendix C).

pair of adjacent layers in the model. Specifically, for a swap intervention at layer $\ell$, we execute the transformer block (including the attention layer, MLP, and normalization) $\ell + 1$ before executing block $\ell$. We record the Kullback-Leibler (KL) divergence between the intervened and original models output distribution, along with the loss, top-1 prediction accuracy, prediction entropy, and benchmark task performance. This intervention allows us to examine how the order of computation affects the model's behavior and performance at different depths.

**Ablation Data Collection**  To generate baselines for each layer swap experiment, we perform zero ablations on the corresponding layer while collecting the same metrics. The ablation preserves the swap ordering: for a swap ordering of **1-2-4**-3-5, the ablation maintains **1-2-4**-5. We opt for zero ablation as opposed to mean ablation, as proposed by [5], to maintain consistency with the swap order.

## 4 Robustness

### 4.1 Intervention Results

We apply our aforementioned drop and swap interventions to every layer of four GPT-2 models [53] and four Pythia models [46]. In Figure 5, we report (1) the KL divergence between the prediction of the intervened model and the nominal model, (2) the fraction of predictions that are the same between the intervened model and the baseline model (denoted as relative accuracy). We also report the performance on common benchmark tasks (HellaSwag[54], ARC-Easy[55] and LAMBADA[56]) for all models in Figure 15-16, which show a similar trend.

In contrast to the first and last layers' interventions, the middle layers are remarkably robust to both deletion and minor order changes. When zooming in on the differences between the effect of swaps and drops for intermediate layers, we find that swapping adjacent layers is less harmful than ablating layers, matching a result in vision transformers [26]. We take this as an indication that certain operations within the forward pass are commutative, though further experimentation is required.

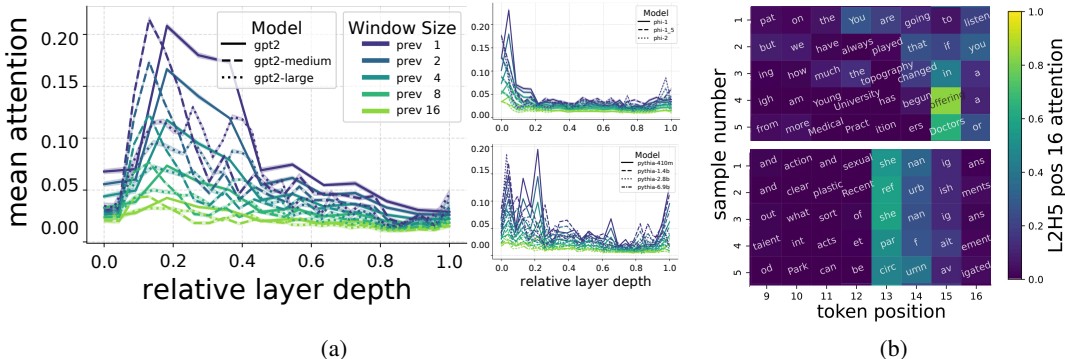

Figure 6: (a) The average (across heads within a layer and query tokens) attention weight placed on the preceding 1, 2, 4, 8, 16 tokens for each layer. (b) Attention from the source token to the final token in various inputs. An identified sub-word merging attention head (bottom) found in the early layers of language models is responsible for attending to multi-token words (i.e, shenanigans, refurbishments, parfaitement, circumnavigate), compared to the baseline set of random non-multi-token words (top).

Intervening on the first layer is catastrophic for model performance for every model, regardless of size or model family. Specifically, dropping or swapping the first layer causes the model to have very high entropy predictions as opposed to causing a mode collapse on a constant token. In some models, swapping the last layer with the second-to-last layer also has a similar catastrophic high-entropy effect, while GPT-2 models largely preserve their predictions. This phenomenon motivates our study into the first few layers of the model, specifically the role paid by attention heads in these layers.

## 5  Stages of Inference Hypothesis

Motivated by the distinct phenomena at the first few and final few layers, we measured representational similarity across each layer output using Centered Kernel Analysis (CKA)[57–59]. This revealed a block-like structure across multiple models as shown in Figure 4. The existence of blocks reflects the robustness observed in the layer-wise intervention. Furthermore, the depth-dependent phase structure indicates that a shared computation motif across adjacent layers occurs in stages.

### 5.1  Stage 1: Detokenization

Given the extreme sensitivity of the model to first-layer ablations, we infer that the first layer is not a normal layer, but rather an extension of the embedding. Uniquely, the first layer is the layer that moves from the embedding basis to that of the transformer's residual stream. It is *only* a function of the current token. Consequently, by ablating the first layer, the rest of the network is blind to the immediate context and is thrown off distribution. Immediately after computing this extended embedding, evidence from the literature suggests that the model concatenates nearby tokens that are part of the same underlying word [60, 61] or entity [62] (e.g., a first and last name). This operation integrates local context to transform raw token representations into coherent entities. In this way, the input is "detokenized" [36, 63]. Previous work has shown the existence of neurons that activate for specific $n$-grams [63, 15]. Of course, to accomplish this, there must be attention heads that copy nearby previous tokens into the current token's residual stream.

**Sub-word Merging Heads**    To further examine this detokenization mechanism, we investigated attention heads responsible for constructing multi-token words, known as *sub-word merging* heads [61]. These heads help capture the context of a token for appropriate prediction, thus contributing to the detokenization process. We constructed a dataset with two classes: each consisting of 16 tokens, where in one class, the final 4 tokens form a word. Our analysis identified specific heads in the early layers of models that contribute solely to the construction of these multi-token words. As illustrated in Figure 6b, layer 2 head 5 of Pythia 2.8B moves information from earlier tokens to the final token of the word. The attention heads exhibit a consistent pattern, where attention decreases as tokens approach the final word. Specifically, the final token of the word attends most strongly to the first

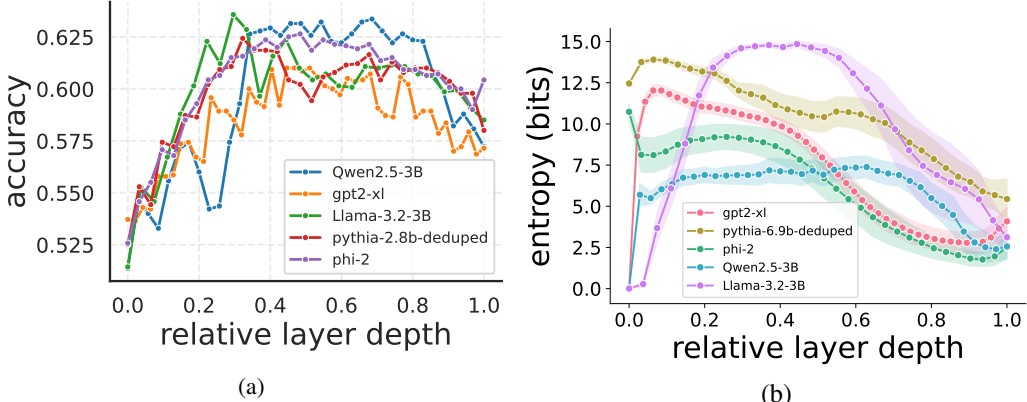

Figure 7: (a) Layer-wise probe accuracy on contextual lexical meaning (WiC task), peaking in intermediate layers is suggestive of where semantic features are linearly encoded. (b) Using the logit lens technique [35], we calculate the probability distribution of the next token at the end of every layer, and then take its entropy. This provides a measure of the model's confidence in the next prediction. Despite high probe accuracy, the residual, but high entropy residual stream suggests that semantic features exist mid-model but are not yet used for prediction. For all models see Appendix 18 and 19.

token, a feature absent in the baseline. This suggests at least one of many mechanisms by which models integrate local context, occurring at higher density in the first half of the models.

**Local Attention** If early layers indeed specialize in integrating local context, then we would expect attention heads in these layers to disproportionately focus on tokens close to the current position. To investigate this hypothesis, we measure the fraction of attention that each token directs toward preceding tokens at varying distances. As shown in Figure 6, attention heads in early layers are strongly biased towards nearby tokens, with attention becoming progressively less localized in deeper layers.

## 5.2 Stage 2: Feature Engineering

After integrating local context in the early layers—e.g., stitching together sub-word tokens and forming short-range dependencies—the model must begin converting those localized representations into more semantically meaningful features. We hypothesize that this marks the beginning of a feature engineering stage, in which the model constructs intermediate features that encode abstract properties useful for downstream prediction.

Prior work provides indirect support for this idea. Model editing studies suggest that factual information is stored in mid-layer MLPs [16, 64, 62], while probing experiments have found that intermediate layers encode features related to sentiment [65], truth [37], and temporal structure [66]. These studies typically show that probing accuracy rises through the early layers, peaks near the midpoint, and then declines, suggesting that features are constructed and later transformed or compressed. Related work also observes a shift from syntactic to semantic representations with depth [36, 38].

**WiC Probing** To illustrate this pattern, we train linear probes to detect context-dependent word meaning using the WiC (Word-in-Context) task [67, 68]. For instance, given two sentences containing the word bank, the task is to classify whether it is used with the same meaning. Examples include distinguishing "the river *bank*" from "the robbed *bank*," where the same word has different meanings depending on the context. We apply this probe at each layer of the model, using the hidden state of the target word in context. As shown in Figure 7 (left), the accuracy of the probe increases through the early layers, peaks in the middle of the model, and then decreases, supporting the hypothesis that semantic features are most *linearly* accessible in the intermediate layers. We extend the observation across model families and sizes in Figure 18.

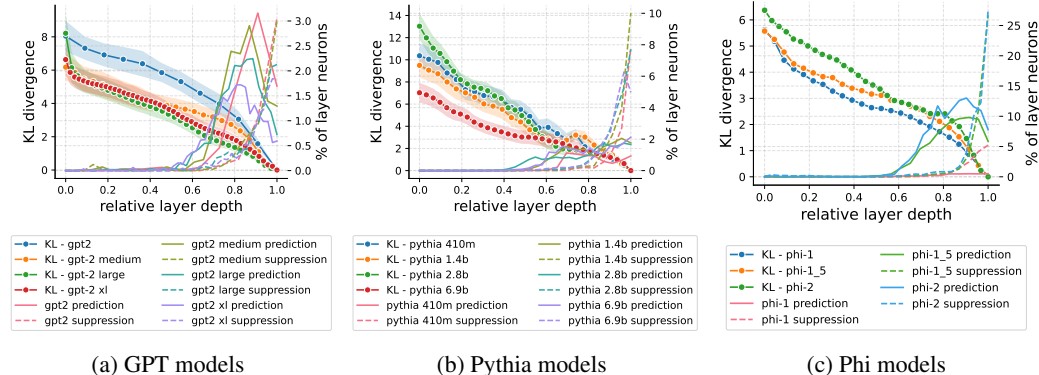

|  | (a) GPT models | (b) Pythia models | (c) Phi models |

Figure 8: We measure KL divergence between intermediate and final predictions using the logit lens method [35]. On the second axis, we use an automated procedure for classifying neuron types detailed in [14], into prediction neurons and suppression neurons. These are universal neurons in all models known to increase the probabilities of tokens and decrease the probabilities of others. We hypothesize this inverse relationship as evidence for ensembling in networks[15].

**Logit Lens** While these results suggest that intermediate representations encode semantic information, it remains unclear whether such features contribute to prediction at this stage. To investigate this, we apply the logit lens [35, 69], which projects the residual stream at each layer into the output vocabulary space using the model's unembedding matrix. This provides a layer-wise estimate of the model's next-token distribution.

We compute both the entropy of the intermediate predictions and their KL divergence from the model output. As shown in Figure 7 (right), entropy remains high and KL divergence low throughout the early and middle layers. In other words, while meaningful features appear to be present in the residual stream at this stage, the model's output distribution remains high in entropy, indicating that these features have not yet been consolidated into confident next-token predictions. Bridging this gap requires a mechanism that selectively retains information from relevant features while filtering out irrelevant ones, thereby reducing uncertainty in the output distribution.

### 5.3 Stage 3: Prediction Ensembling

Around the midpoint of the model, we observe a qualitative shift in behavior. Having constructed semantic features in the earlier layers, the model must begin converting these into specific next-token predictions. Evidence for this transition comes from the logit lens, where we observe a steady decline in entropy (Figure 7 right) and KL divergence (Figure 8) between intermediate and final predictions beginning around the middle layers. This suggests that the model is gradually committing to a particular output, aggregating semantic features into a more concrete distribution over tokens.

This region of the model also displays high robustness to layer interventions (Figure 5), suggesting redundancy or capacity for self-repair. One possible cause of this resilience is the presence of overlapping computational pathways [6, 70]. Rather than relying on a single deterministic path, the model seems to combine multiple signals—both across and within layers—to form its prediction. We explore this mechanism by identifying the neurons that contribute to prediction, testing their collective behavior through a case study, and analyzing their distributional effects across depth.

**Ensembling** Within these overlapping pathways, we investigate specialized ensembles known as prediction neurons—units that systematically promote the likelihood of specific tokens [15, 7, 14]. These neurons work in tandem with suppression neurons (discussed in Section 5.4) to shape the model's output.

**Prediction and Suppression neurons** Following previous work[14], we identify these neurons by analyzing the MLP output weights $\mathbf{w}_{out}$ and their projection into vocabulary space via the unembedding matrix $\mathbf{W}_U$. Prediction neurons exhibit a logit effect distribution $\mathbf{W}_U \cdot \mathbf{w}_{out}$ with high kurtosis and positive skew; suppression neurons exhibit high kurtosis and negative skew. Across 18

models, prediction neurons begin to appear around the midpoint, increasing in density through the latter layers (Figure 8), before being overtaken by suppression neurons. For a detailed analysis of the detection and characterization of prediction, suppression, and other "universal" neurons, we refer readers to the original work [14].

**Probing for the Suffix "-ing"**  We hypothesize that ensembles of prediction and suppression neurons collectively support next-token prediction. To test this, we construct a balanced classification task: given a 24-token context to a verb, does the final token end with or without "-*ing*"? We train linear probes on the activations of 32 high-variance prediction and suppression neurons, both individually and in groups. Neurons are selected using the criteria above, and examples from GPT-2 XL are shown in Figure 9. The full neuron list appears in Appendix 21.

We train two types of probes on the penultimate token's activations: 32 individual neuron probes and top-$k$ ensemble probes ranked by individual accuracy (Figure 9). Suppression neurons yield the strongest individual probes, performing on par with the model's predictions (dotted red line). Ensemble probes trained on prediction neurons outperform both individual neurons and the model average, suggesting an important interplay between the two neuron types.

**Density Effects**  The balance between prediction and suppression neurons appears to shape the model's output. To test this, we analyze how their density relates to the KL divergence between each layer's logit lens distribution and the final output. The sharpest decline in divergence corresponds closely with the rise in prediction neuron density, which peaks at roughly 85% of model depth.

Model comparisons further reinforce this pattern. Phi-1 has fewer prediction neurons and a shallower KL slope compared to later Phi models (Figure 8c). GPT and newer Phi models show steeper, smoother KL divergence drops than Pythia (Figures 8a, 8b). Notably, the most performant Phi models exhibit nearly 15% prediction and 25% suppression neurons per layer—5–8× the density in GPT-2 and 3–7× that of Pythia.

Interestingly, the density of prediction neurons decreases near the final 10% of layers, even as the model continues to converge on its output, sometimes accelerating(Figure 8b). This suggests the involvement of a distinct final-stage mechanism, which we delineate as a separate stage.

## 5.4  Stage 4: Residual Calibration

As prediction neuron density declines in the final layers, a different mechanism emerges. Across all models, we observe a sharp rise in suppression neurons near the end of the network. This transition from prediction to suppression neurons frequently coincides with an inflection point: entropy stops decreasing and begins increasing in the final layers (Figure 19b). Unlike prediction neurons, which promote likely tokens, suppression neurons refine the model's output by removing obsolete features and down-weighting improbable tokens. The resulting entropy increase in the final layers suggests that suppression neurons serve to calibrate the model's output toward the task it was trained for: producing a well-formed distribution over possible next tokens.

**Layer Repeating Experiment**  To further explore this hypothesis, we design an experiment where we *repeat* certain layers of the model. Specifically, we duplicate blocks of layers within the model—for example, repeating layers 5 through 7 results in a sequence like (...4-5-5-6-6-7-7-8-9...). For this analysis, we fix the number of repeats to 1 and the block length to 5 (see additional results across model sizes and block length in Figure 25,23). In Figure 11, we observe that repeating blocks in the latter half of the model leads to a consistent decrease in entropy relative to the baseline (horizontal line). When evaluated on downstream benchmarks, the models with repeated layers at the last 80-90% of depth also exhibit improved performance on benchmarks, suggestive of residual calibration and the late-stage influence of prediction and suppression neurons. (Appendix 24).

**Final Layer**  The intensity of suppression neurons, as seen in Figure 8, is localized in the final few layers of the model, where the quantity of suppression neurons outstrips prediction neurons. To quantify the *intensity* of this change to the output distribution, we measure the norm of the MLP output, where a larger norm suggests a greater contribution to the residual (Figure 10). This also coincides with an increase in entropy (Figure 19b).

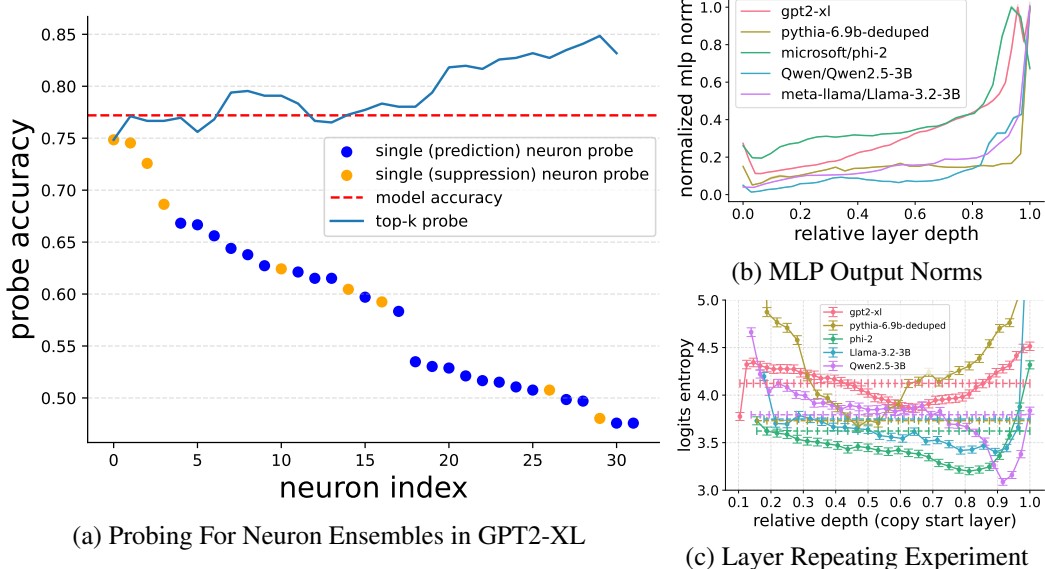

(a) Probing For Neuron Ensembles in GPT2-XL

(b) MLP Output Norms

(c) Layer Repeating Experiment

Figure 12: (a) Accuracy of linear probes trained to predict whether the final token ends in "-ing," using activations from individual prediction and suppression neurons (scatter points) and ensembles of neurons (blue line). Ensembles outperform individual probes and occasionally exceed the model's top-1 accuracy (red dotted line), consistent with the presence of "prediction ensembling." (b) Layer-wise MLP output norms across all 18 models show a rise toward the final layers, suggesting increasing residual contribution late in the model. (c) Repeating layers from the later half of a model reduces final-layer logit entropy more than repeating earlier layers or using the original model (dotted line), suggestive of residual calibration and the late-stage influence of prediction and suppression neurons.

## 6   Concluding Remarks

**Why Are Language Models Robust to Layer-Wise Interventions?**   We hypothesize that the robustness of language models to layer deletion and swapping stems in part from the transformer's residual architecture. This interpretation aligns with our findings on prediction and suppression neurons: multiple computational pathways appear to contribute to the same output, allowing the network to tolerate disruption in any single path. The residual stream promotes this "ensembling", enabling gradient descent to construct shallow sub-networks that can operate in parallel. This architectural flexibility reduces the model's reliance on any specific layer, explaining its resilience to local interventions and supporting observed self-repair behavior and overlapping representations.

**Limitations and Future Work**   While our four-stage hypothesis captures broad, depth-dependent patterns in LLMs, several caveats remain. Stage boundaries are approximate, and multiple stages may co-occur within a single layer. The findings reflect aggregate trends, whereas individual tokens may follow distinct processing paths. Additionally, we do not isolate the factors behind model-specific differences; e.g., GPT's greater robustness could arise from dropout, architectural variations, or depth. These limitations point directly to promising directions for future research. Future work should seek to sharpen these boundaries, link them to optimization dynamics, and test this hypothesis with a theoretical account to explain the empirical results.

**Conclusion**   This work introduces a four-stage hypothesis for understanding inference in large language models, grounded in a diverse set of behavioral and mechanistic analyses. By examining how models respond to structural interventions—layer deletion and swapping—as well as probing attention patterns, neuron function, and residual stream dynamics, we identify a repeatable depth-wise structure to model computation. These stages—detokenization, feature engineering, prediction ensembling, and residual calibration—emerge across architectures and scales, suggesting that transformers perform inference not as a flat pipeline but as an ordered composition of specialized computational regimes. While not definitive, the strength and consistency of the empirical signatures presented here provide compelling evidence in support of the proposed hypothesis. Rather than aiming for an exhaustive

mechanistic dissection, we offer a unifying perspective that synthesizes and extends prior findings in and outside the interpretability literature.

More broadly, this layered view of inference has implications for how we interpret, audit, and intervene on language models. Understanding not just what a model computes, but when and where it computes it, may inform future approaches to alignment, compression, and modularity in model design. We hope this hypothesis serves as a foundation for a deeper investigation into the emerging capabilities of LLMs.

**Contributions and Acknowledgments**   VL conceived and led the study, performed the analyses, and drafted the paper. JL conducted the sharpening experiment, WiC analysis, benchmarking, and CKA experiments. JL, WG and MT contributed to experimental design and analytical methodology, provided critical revisions, and WG and JL assisted with writing. A big thanks to Katrin Franke, Surya Ganguli, Sophia Sanborn, Eric Michaud, Josh Engels, Dowon Baek, Isaac Liao for helpful feedback.

We want to extend a special acknowledgment to Elias Sandmann, who notified us of an error in the Transformer Lens documentation that affected our implementation of Logit Lens. The correction was crucial for the accuracy and reproducibility of our results which may have gone overlooked without his concern.

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

## A Experiment Diagram

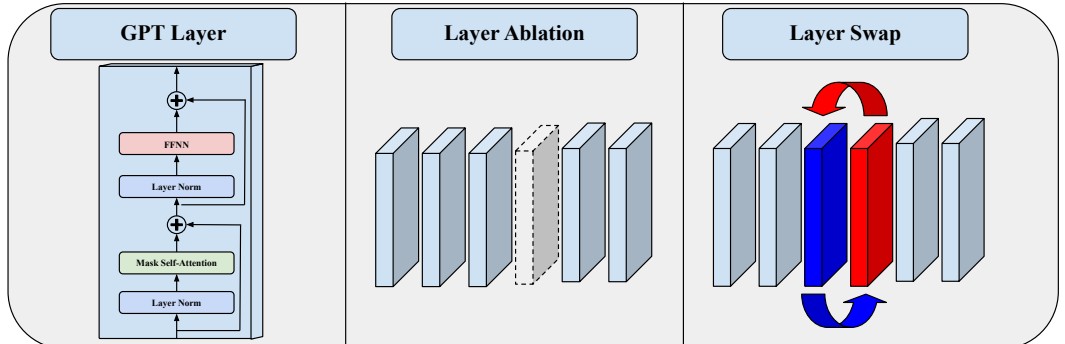

Figure 13: To study the stages of inference, we perform two experiments, each a layer-wise intervention, where a layer (left) encompasses all model components. The first intervention is a zero ablation (i.e, layer removal) of the layer (middle), in which a layer is fully removed and residual connections skip the layer entirely. The second intervention (last) is an adjacent layer swap, in which we permute the positions of two layers. The ablation is performed on all layers, while the layer swap is performed on all adjacent pairs of layers in the model.

| Name | HuggingFace Model Name |
|------|------------------------|
| Pythia 410M | EleutherAI/pythia-410m-deduped |
| Pythia 1.4B | EleutherAI/pythia-1.4b-deduped |
| Pythia 2.8B | EleutherAI/pythia-2.8b-deduped |
| Pythia 6.9B | EleutherAI/pythia-6.9b-deduped |
| Pythia 12B | EleutherAI/pythia-12b-deduped |
| GPT-2 Small (124M) | gpt2 |
| GPT-2 Medium (355M) | gpt2-medium |
| GPT-2 Large (774M) | gpt2-large |
| GPT-2 XL (1.5B) | gpt2-xl |
| Phi 1 (1.3B) | microsoft/Phi-1 |
| Phi 1.5 (1.3B) | microsoft/Phi-1.5 |
| Phi 2 (2.7B) | microsoft/Phi-2 |
| Qwen 0.5B | Qwen/Qwen2.5-0.5B |
| Qwen 1.5B | Qwen/Qwen2.5-1.5B |
| Qwen 3B | Qwen/Qwen2.5-3B |
| Qwen 14B | Qwen/Qwen2.5-14B |
| Llama-3.2 1B | meta-llama/Llama-3.2-1B |
| Llama-3.2 3B | meta-llama/Llama-3.2-3B |
| The Pile | EleutherAI/the_pile_deduplicated |

Table 3: List of models and dataset used in the experiments.

## B Additional Empirical Details

**Github**   All experimental code for future experiments is available at:
https://github.com/vdlad/Remarkable-Robustness-of-LLMs.

**Transformer Lens**   We make ubiquitous use of Transformer Lens [71] to perform hooks and transformer manipulations.

**HuggingFace**   For specificity, we utilize the following HuggingFace model names, and dataset. We do not change the parameters of the models from what they are described on the HuggingFace page.

All experiments described can be performed on a single NVIDIA A6000. We utilized 2 NVIDIA A6000 and 500 GB of RAM. To aggregate the metrics described in the paper, we run the model on 1

million tokens $\ell$ times, where $\ell$ is the number of layers. This takes on average 8 hours per model, per layer intervention (swapping and ablating). We save this aggregation for data analysis.

**Residual Sharpening to Residual Calibration**    We initially named the final stage of inference "residual sharpening" but have renamed it to "residual calibration" for greater accuracy. While suppression neurons do eliminate obsolete features from the model's representation, the final layers sometimes exhibit an increase in entropy—a seemingly contradictory behavior if the goal were simply to sharpen predictions toward a single top token. Instead, this entropy increase suggests that suppression neurons calibrate the representation to produce a well-formed distribution over possible next tokens, aligning with the language modeling objective. This calibration process differs from sharpening, which would imply converging toward a single prediction. Additionally, models even within the same family exhibit varying entropy patterns in their final layers. We hypothesize that the variation in entropy of the final layers may indicate model confidence and contribute to hallucination behavior, though we leave this investigation to future work.

**Layer Norm Preprocessing**    We utilize several conventional weight preprocessing techniques to streamline our calculations [71].

Following [14], before each MLP calculation, a layer norm operation is applied to the residual stream. This normalizes the input before the MLP. The TransformerLens package simplifies this process by incorporating the layer norm into the weights and biases of the MLP, resulting in matrices $W_{\text{eff}}$ and $b_{\text{eff}}$. In many layer norm implementations, trainable parameters $\boldsymbol{\gamma} \in \mathbb{R}^n$ and $\mathbf{b} \in \mathbb{R}^n$ are included:

$$\text{LayerNorm}(\mathbf{x}) = \frac{\mathbf{x} - \mathbb{E}(\mathbf{x})}{\sqrt{\text{Var}(\mathbf{x})}} * \boldsymbol{\gamma} + \mathbf{b}. \tag{1}$$

We "fold" the layer norm parameters into $W_{\text{in}}$ by treating the layer norm as a linear layer and then merging the subsequent layers:

$$\mathbf{W}_{\text{eff}} = \mathbf{W}_{\text{in}} \, \mathbf{diag}(\boldsymbol{\gamma}) \qquad \mathbf{b}_{\text{eff}} = \mathbf{b}_{\text{in}} + \mathbf{W}_{\text{in}}\mathbf{b} \tag{2}$$

Additionally, we then center reading weights. Thus, we adjust the weights $\mathbf{W}_{\text{eff}}$ as follows:

$$\mathbf{W}^{'}_{\text{eff}}(i, :) = \mathbf{W}_{\text{eff}}(i, :) - \bar{\mathbf{W}}_{\text{eff}}(i, :)$$

**Centering Writing Weights**    Because of the LayerNorm operation in every layer, we can align weights with the all-one direction in the residual stream as they do not influence the model's calculations. Therefore, we mean-center $\mathbf{W}_{\text{out}}$ and $\mathbf{b}_{\text{out}}$ by subtracting the column means of $\mathbf{W}_{\text{out}}$:

$$\mathbf{W}^{'}_{\text{out}}(:, i) = \mathbf{W}_{\text{out}}(:, i) - \bar{\mathbf{W}}_{\text{out}}(:, i)$$

**Extension of Results in Larger Models (>10B parameters)**    In Appendix K, we extend the results of the core experiments in larger models with more than 10B parameters (Qwen2.5-14B, Pythia-12B).

**Societal Impact**    We do not anticipate any immediate societal impact from this research.

# C Centered Kernel Alignment (CKA)

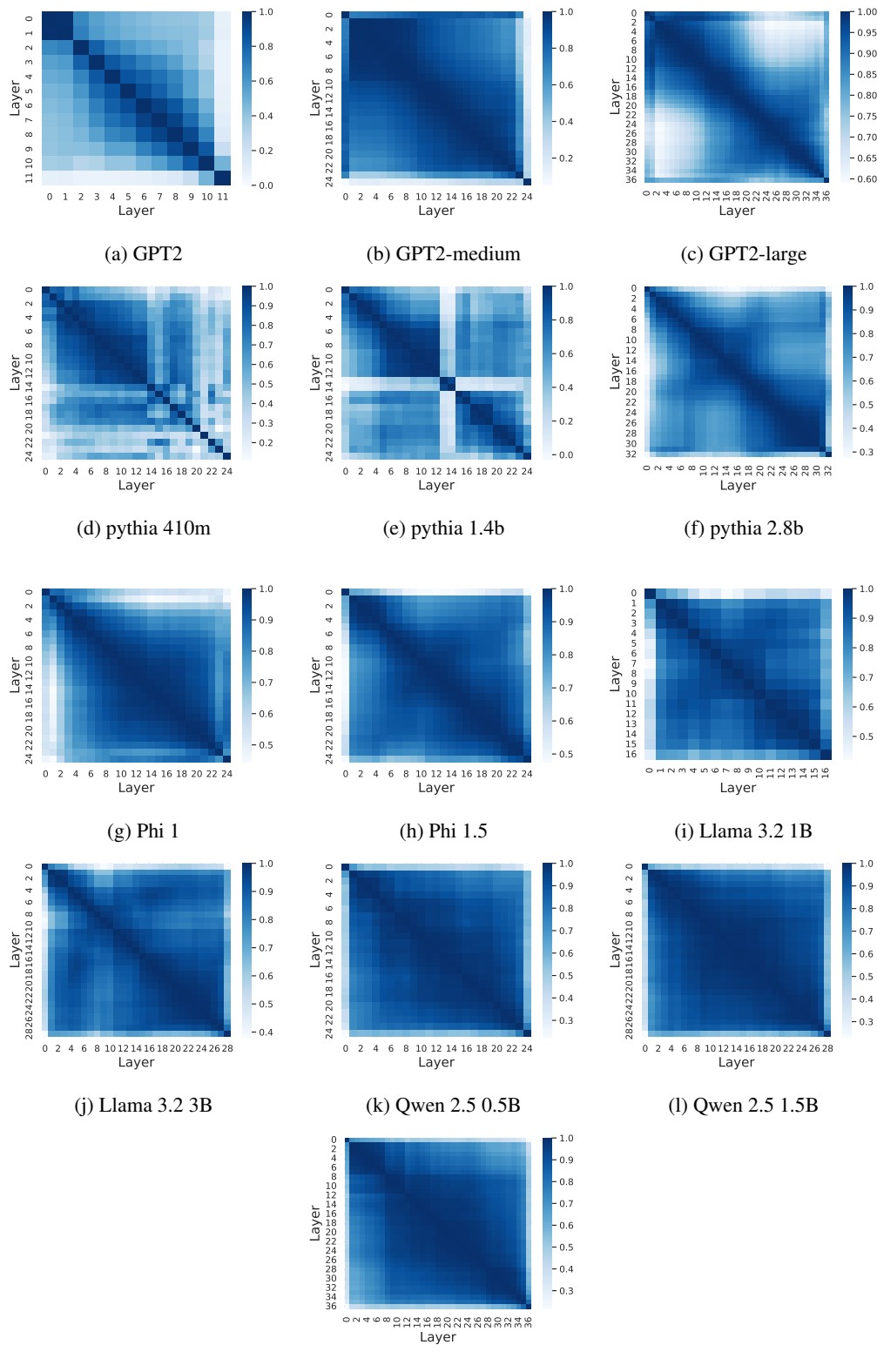

(a) GPT2

(b) GPT2-medium

(c) GPT2-large

(d) pythia 410m

(e) pythia 1.4b

(f) pythia 2.8b

(g) Phi 1

(h) Phi 1.5

(i) Llama 3.2 1B

(j) Llama 3.2 3B

(k) Qwen 2.5 0.5B

(l) Qwen 2.5 1.5B

(m) Qwen 2.5 3B

Figure 14: CKA across layers from the last token representation sampled from Pile dataset (max token length 512, batch size 128). We used unbiased CKA [72, 59].

# D  Benchmark Tasks Performance After Layer-Wise Intervention

We evaluate the benchmark performance on HellaSwag, ARC-Easy and LAMBADA [54–56] with the intervened model. We observe a similar trend to KL divergence reported in the main paper. Generally, the intervention at the first layer and the last layer shows catastrophic deterioration of the performance but intervention on intermediate layers shows robust performance.

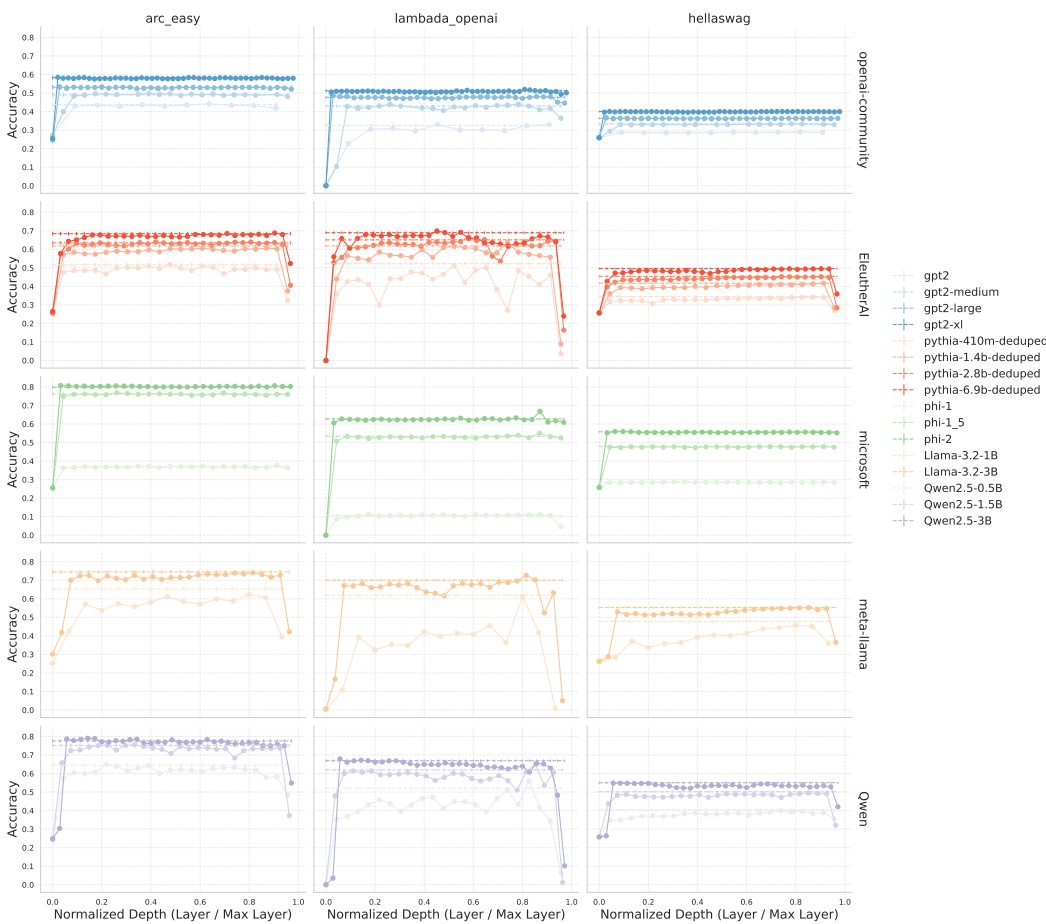

Figure 15: Benchmark task performance after layer swap. Baseline performance of each model is marked with a dotted horizontal line.

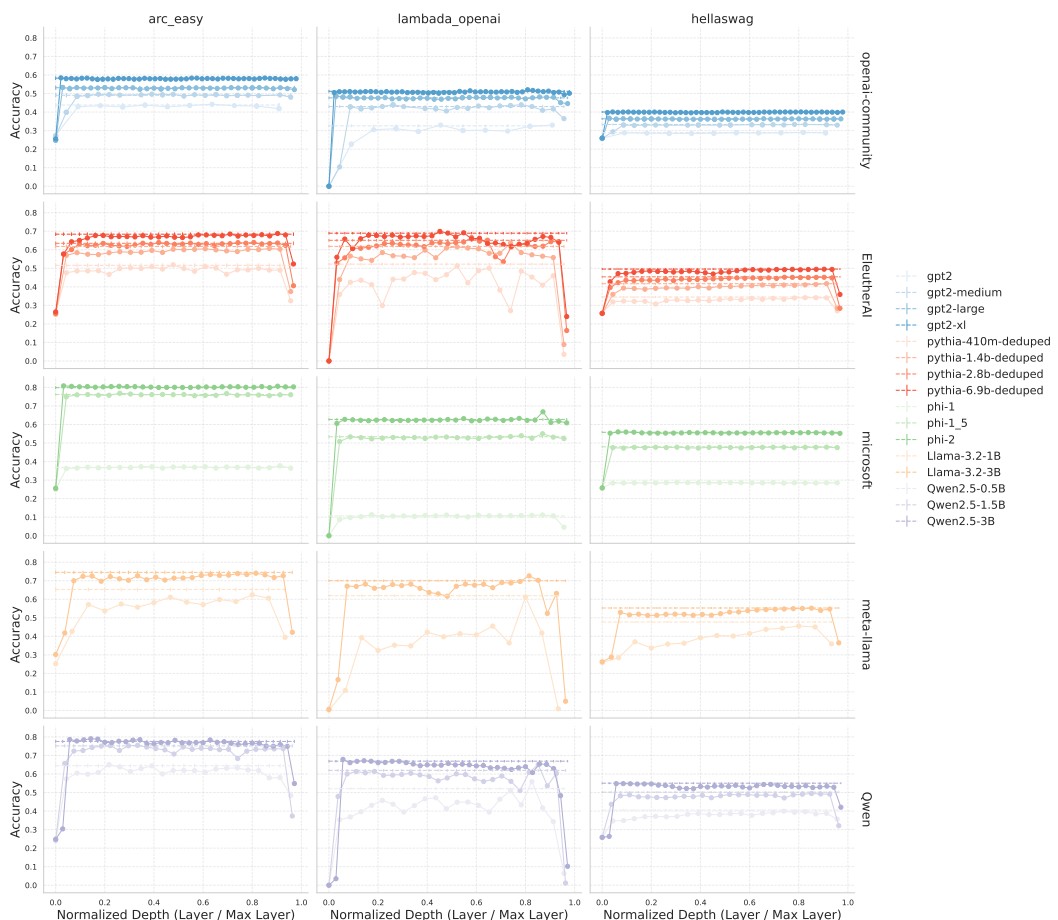

Figure 16: Benchmark task performance after layer swap. Baseline performance of each model is marked with a dotted horizontal line.

# E   Prediction and Suppression Neuron

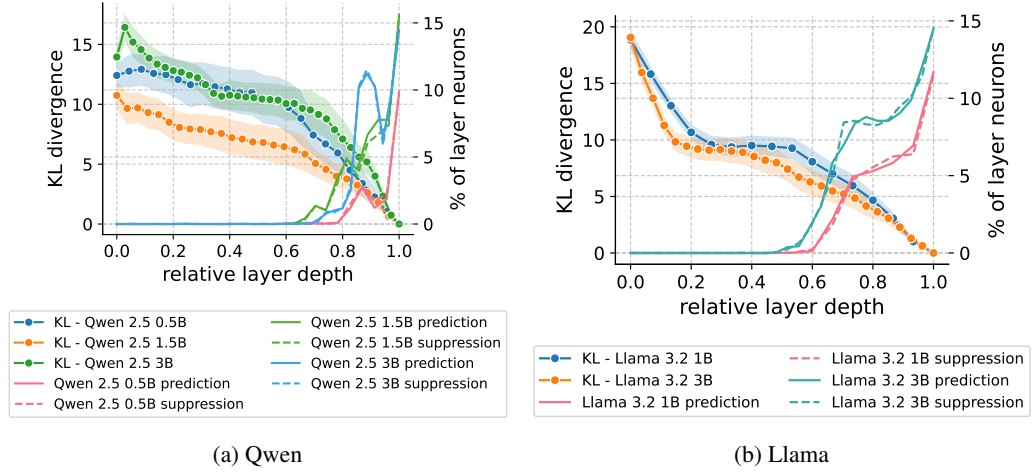

(a) Qwen

(b) Llama

Figure 17: Prediction and Suppression neurons for Qwen and Pythia.

# F   WiC contextual word probe

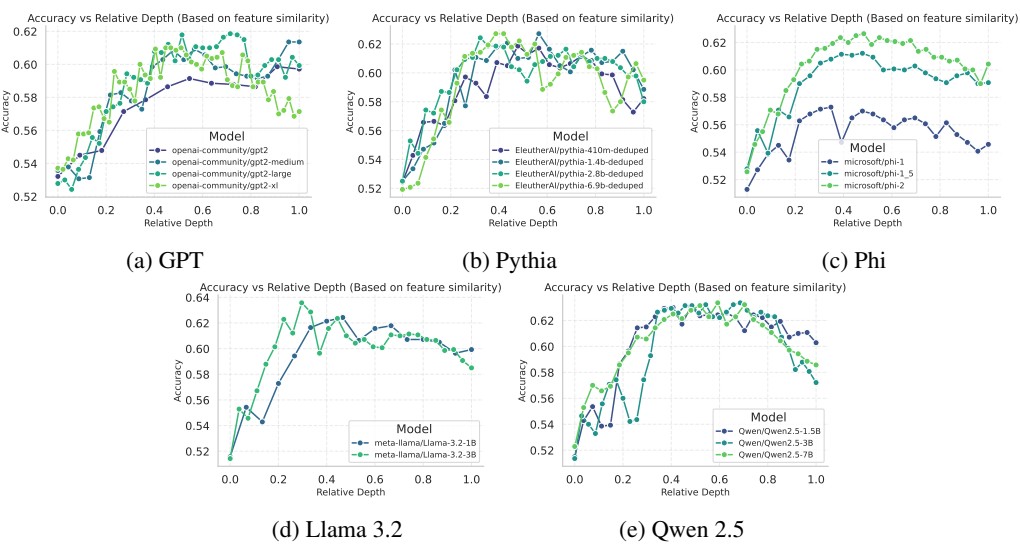

(a) GPT          (b) Pythia          (c) Phi

(d) Llama 3.2          (e) Qwen 2.5

Figure 18: WiC probing accuracy over layers across model families and sizes. Across all models and sizes, we observe the probe accuracy related to contextual semantics of lexical items gradually increases and peaks around the middle layers and degrades.

# G   Logit Lens Entropy

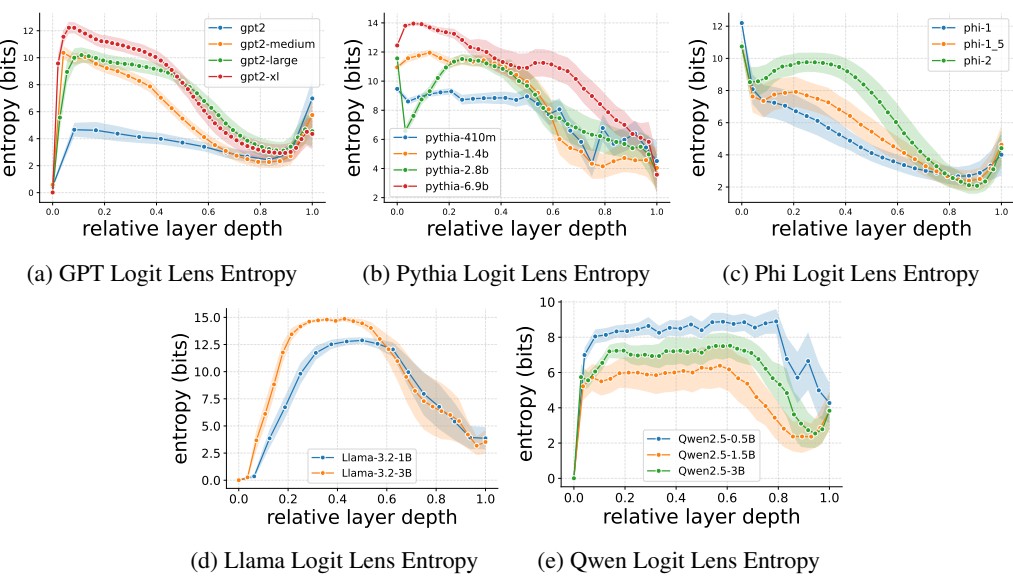

(a) GPT Logit Lens Entropy     (b) Pythia Logit Lens Entropy     (c) Phi Logit Lens Entropy

(d) Llama Logit Lens Entropy     (e) Qwen Logit Lens Entropy

Figure 19: Using the logit lens technique [35], we calculate the probability distribution of the next token at the end of every layer, and then take its entropy.

# H   MLP Norms

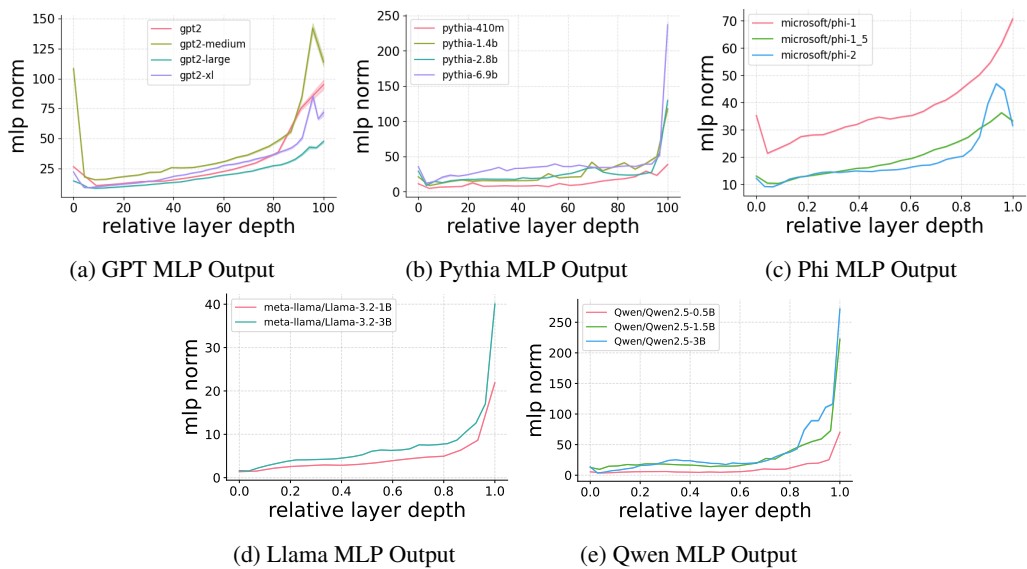

(a) GPT MLP Output

(b) Pythia MLP Output

(c) Phi MLP Output

(d) Llama MLP Output

(e) Qwen MLP Output

Figure 20: The norm of the output of every MLP across its layers to measure its contribution to the residual stream. Across all 18 models, the norm grows and peaks in the final layers before output, suggestive of the final two stages of inference, predictive ensembling, and residual calibration

# I Top Prediction and Suppression Neurons

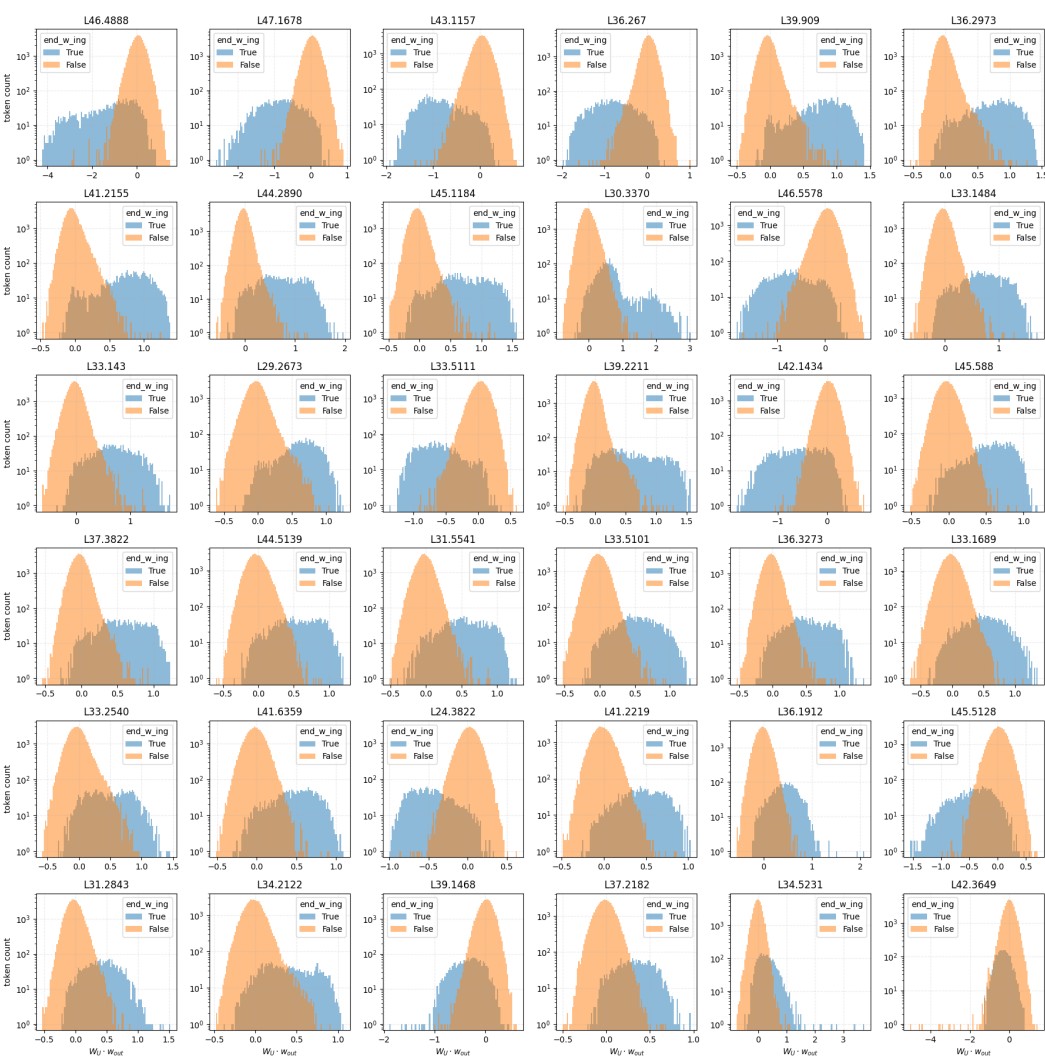

Figure 21: Top 36 prediction and suppression neurons for -ing which have the greatest mean absolute difference between respective ($W_U \cdot w_{\text{out}}$). Elements with a negative skew are suppression neurons for the respective labeled class, while elements with a positive skew are prediction neurons. This is calculated by calculating the product between the model unembedding weights and output weights of MLP.

# J    Layer repeats experiment

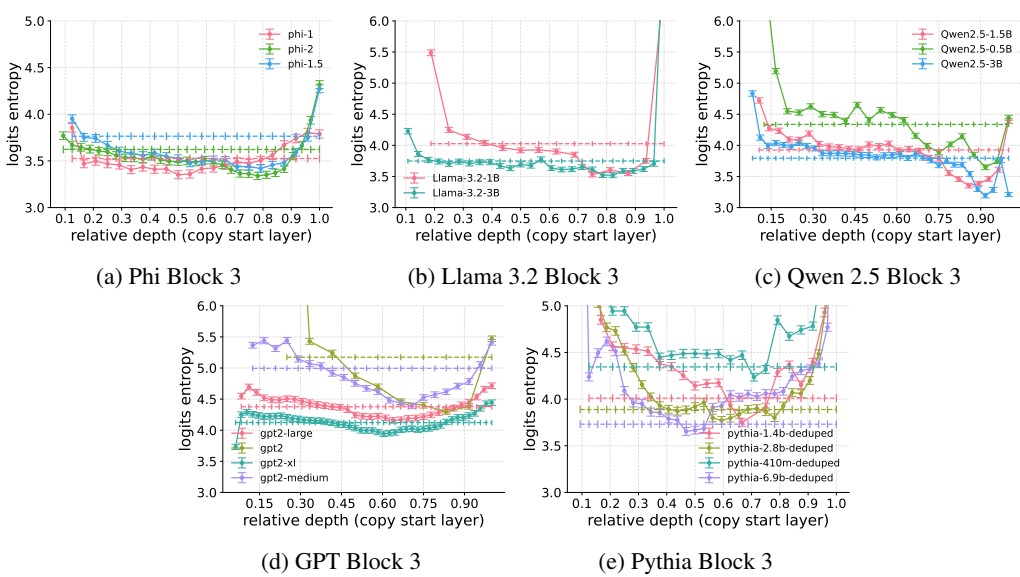

(a) Phi Block 3

(b) Llama 3.2 Block 3

(c) Qwen 2.5 Block 3

(d) GPT Block 3

(e) Pythia Block 3

Figure 22: Block 3 repeat experiment.

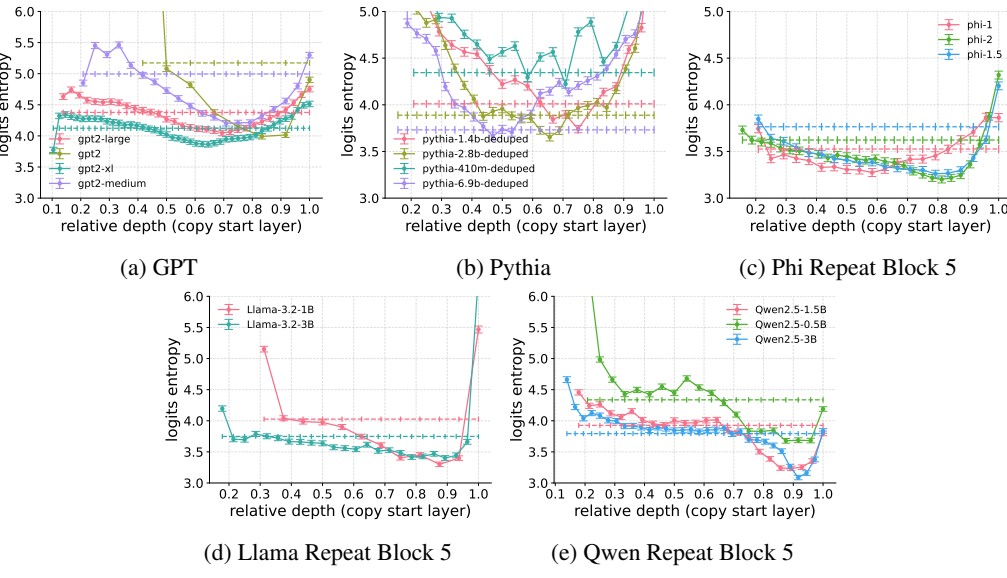

(a) GPT

(b) Pythia

(c) Phi Repeat Block 5

(d) Llama Repeat Block 5

(e) Qwen Repeat Block 5

Figure 23: Block 5 repeat experiment on additional models.

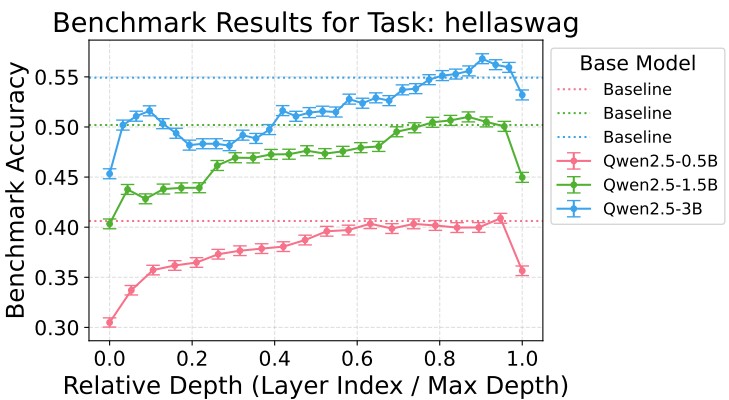

Figure 24: Qwen repeat 5 model's performance on Hellaswag

# K Experiments on Larger Models (>10B parameters)

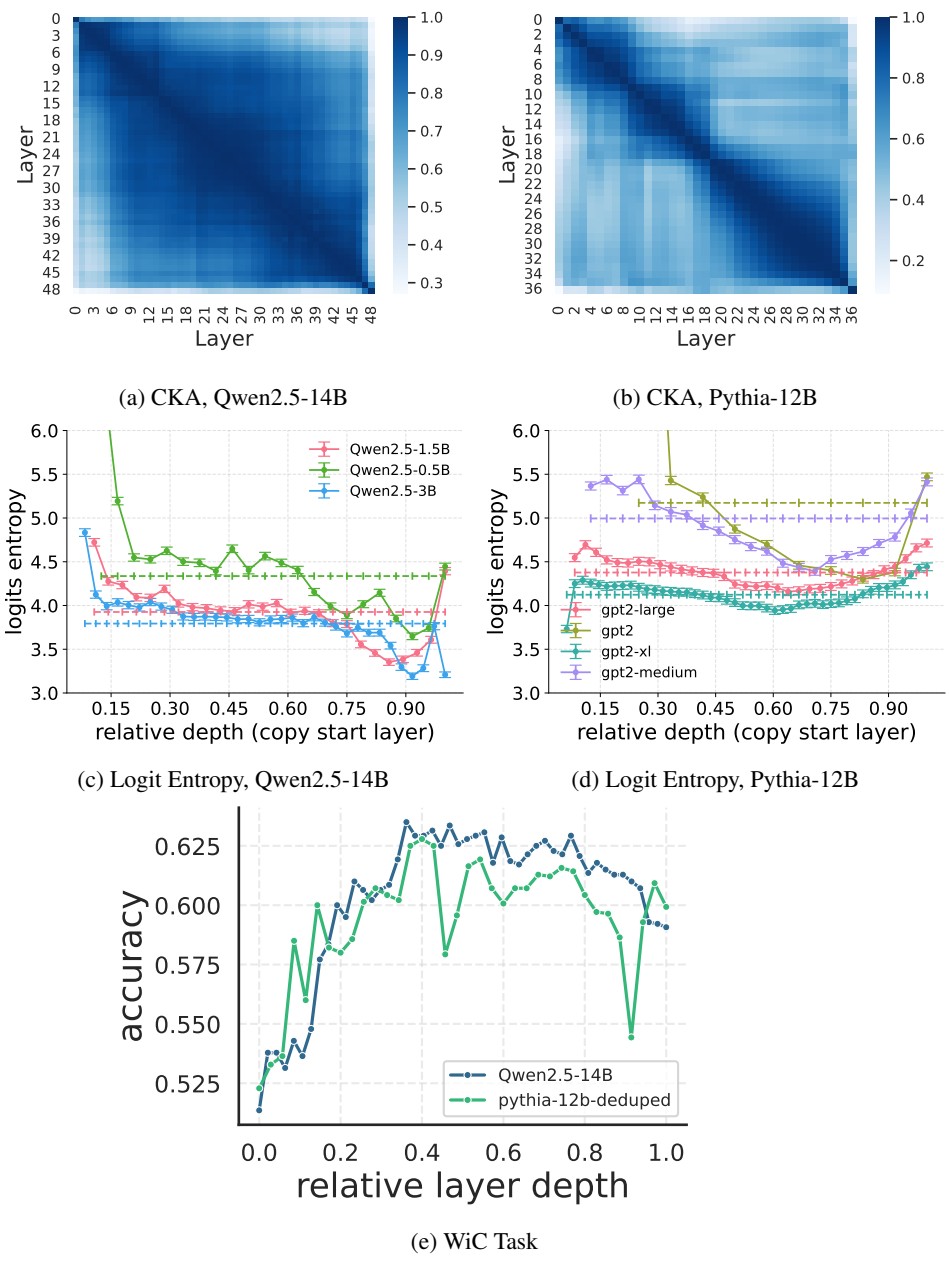

(a) CKA, Qwen2.5-14B

(b) CKA, Pythia-12B

(c) Logit Entropy, Qwen2.5-14B

(d) Logit Entropy, Pythia-12B

(e) WiC Task

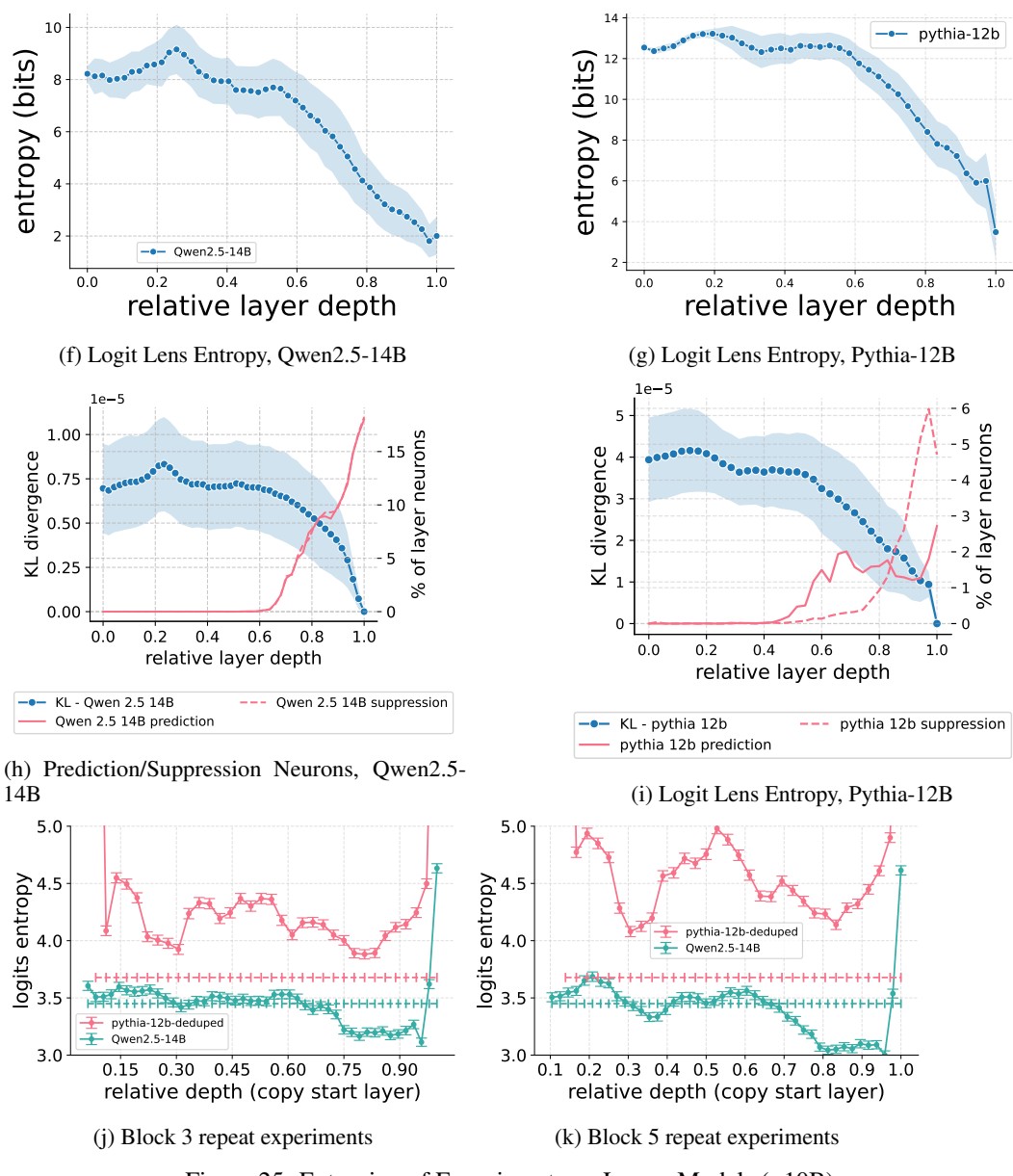

(f) Logit Lens Entropy, Qwen2.5-14B

(g) Logit Lens Entropy, Pythia-12B

(h) Prediction/Suppression Neurons, Qwen2.5-14B

(i) Logit Lens Entropy, Pythia-12B

(j) Block 3 repeat experiments

(k) Block 5 repeat experiments

Figure 25: Extension of Experiments on Larger Models (>10B).

# L    Experiment Subset on OOD Data

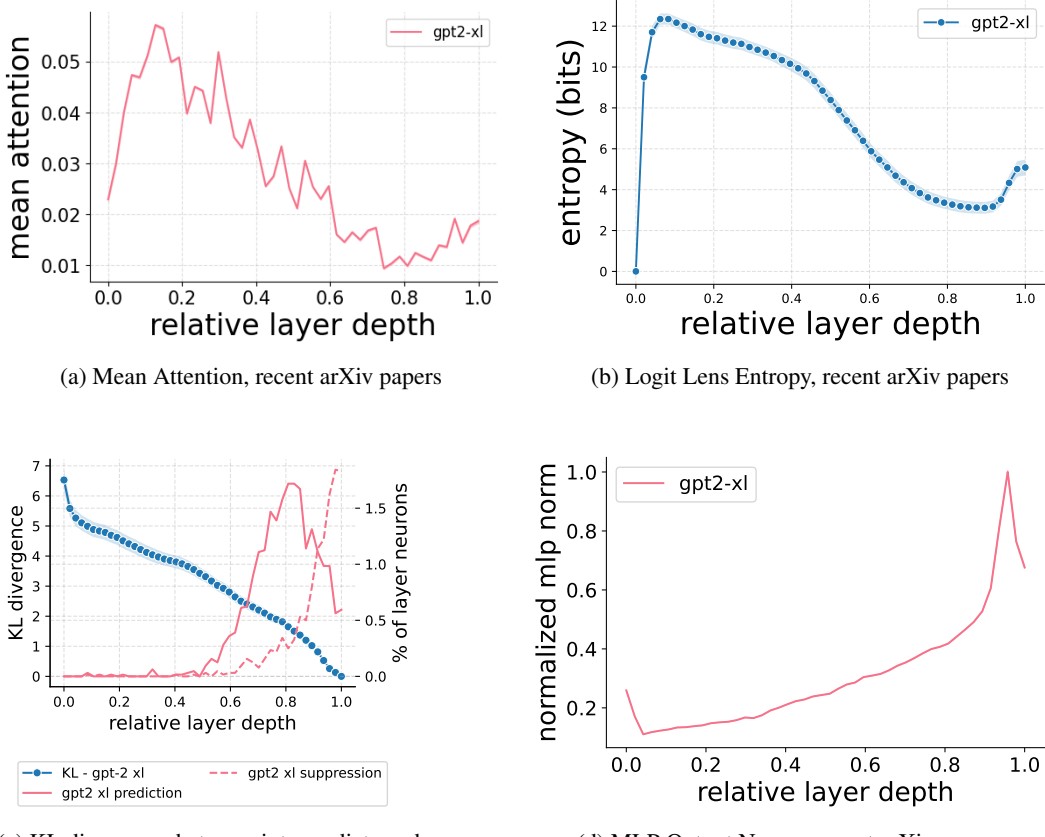

(a) Mean Attention, recent arXiv papers

(b) Logit Lens Entropy, recent arXiv papers

(c) KL divergence between intermediate and output distributions, recent arXiv papers

(d) MLP Output Norms, recent arXiv papers

Figure 26: Running on 2024–2025 arXiv papers, code, and languages results in consistent patterns across the hypothesized stages of inference for GPT-XL.

