# OpenReview forum: "Remarkable Robustness of LLMs: Stages of Inference?"
_NeurIPS.cc/2025/Conference — NeurIPS 2025 poster_

### Official Review · Reviewer_8AAd · 2025-06-29

**Clarity:** 3
**Significance:** 1
**Originality:** 1
**Rating:** 2
**Confidence:** 3

**Summary:**

This paper investigates the robustness of language models to structural changes such as ablating or transposing adjacent layers. Experiments in the paper demonstrate that middle layers are robust to these layer-wise interventions whereas starting and ending layers are not. The authors then use this experimental evidence to hypothesize that language models operate with respect to four stages of inference: detokenization, feature engineering, prediction ensembling, and residual sharpening; some preliminary evidence is gathered to support this hypothesis.

**Questions:**

- What additional insight into the stages of inference of language models is gained in performing layer ablations and layer swapping over existing probing methods?
- Please see my other questions and concerns above.

**Ethical Concerns:**

["NO or VERY MINOR ethics concerns only"]

**Limitations:**

yes

**Quality:**

2

**Strengths And Weaknesses:**

**Strengths**
- This paper addresses the issue of interpreting the inner workings of Transformer language models which is an important problem.
- The results with swapping adjacent layers is interesting and raises questions about the extent to which Transformer operations are commutative.
- The presentation is generally well done, and the paper is easy to follow.


**Major Weaknesses**
- It is hard for me to pin down the exact contributions of the paper. One one hand, much of the insights into the robustness of language models to layer ablations has already been explored extensively in works like [4, 6, 7]. The relationship between a model's circuit and its representations have also been the subject of extensive study through areas such as representation engineering or steering [8, 9]. On the other hand, the stages of inference hypothesis that the authors propose has been sort of known and also has been extensively explored using probing.

- The models examined in the paper are relatively small. It is unclear how these insights would transfer to much larger models with more complex mechanisms.

- The paper dedicates much writing to the "stages of inference." However, there is little evidence to support this claim, especially on downstream tasks.

- All of the experiments are performed on the Pile which presumably is close to the training corpus of these large models. It is unclear how the results in this paper would transfer to more practical settings for the practitioner like in-context learning, instruction following, finetuning, etc.


**Other Weaknesses**

- There are many places in the paper where I believe the literature is misrepresented or need to be discussed in greater detail.

   - > ...mechanistic interpretability, has explored the iterative inference hypothesis...However, this iterative view contrasts with the "circuit" hypothesis... (Lines 19-27)

    I do not see how the iterative inference hypothesis contrasts or contradicts the circuit hypothesis. Take for example the work of [1] which discovers "negative name mover heads" that also "iteratively" add onto the residual stream by suppressing the wrong answer, or the work of [2, 3] which leverage the residual stream to formally discover circuits.

    - > ...Building on these insights, we analyze depth-wise roles and synthesize our findings...to bridge the top-down and bottom-up view of computation in decoder-only LLMs... (Lines 40-44).

     What views are these, why are they different from each other, and why do they need to be bridged? I do not understand how any of the results in the paper bridge any top-down or bottom-up views. Specifically, the paper proceeds in a very bottom-up manner, through data-driven ablations on the model's representations. Then, hypotheses are made about top-level behaviors with little evidence.

    - > Prior work on ablations and layer-wise analysis has primarily focused on BERT... (Lines 246-248)

    In the papers that the authors cite themselves [4, 5] on existing work that performs layer-wise ablations in LMs, large modern decoder-only architectures are studied. For example, [4] examines models like Llama 2 (7b), Baichuan 2 (7b), while [5] looks at Llama 2 (70b). Other than the stages of inference hypothesis, what are the differences between the authors' findings and the findings of these papers?


[1] Interpretability in the Wild: A Circuit for Indirect Object Identification in GPT-2 Small. Kevin Ro Wang, Alexandre Variengien, Arthur Conmy, Buck Shlegeries, Jacob Steinhardt. ICLR 2023.

[2] A Mathematical Framework for Transformer Circuits. Nelson Elhage and others. Transformer Circuits Thread 2021.

[3] Finding Transformer Circuits with Edge Pruning. Adithya Bhaskar, Alexander Wettig, Dan Friedman, Danqi Chen. NeurIPS 2024.

[4] The Unreasonable Ineffectiveness of the Deeper Layers. Andrey Gromov, Kushal Tirusamla, Hassan Shapourian, Paolo Glorioso, Daniel Roberts. ICLR 2025.

[5] ShortGPT: Layers in Large Language Models are More Redundant than You Expect. Xin Men, Mingyu Xu, Qingyu Zhang, Bingning Wang, Hongyu Lin, Yaojie Lu, Xianpei Han, Weipeng Chen.

[6] Eliciting latent predictions from transformers with the tuned lens. Nora Belrose, Zach Furman, Logan Smith, Danny Halawi, Igor Ostrovsky, Lev McKinney, Stella Biderman, Jacob Steinhardt.

[7] Interpreting GPT: The Logit Lens. nostalgebraist. Less Wrong 2020.

[8] Representation Engineering: A Top-Down Approach to AI Transparency. Andy Zou, Long Phan, Sarah Chen, James Campbell, Phillip Guo, Richard Ren, Alexander Pan, and others.

[9] Function vectors in large language models. Eric Todd, Millicent Li, Arnab Sen Sharma, Aaron Mueller, Byron C Wallace, David Bau. ICLR 2024.

---

> ### Author Rebuttal · Authors · 2025-07-31
>
> # Response to Reviewer 5
>
> We thank the reviewer for taking time to review our work. We appreciate that you found our work poses an important question and that the results were well presented.
>
> The major concerns appear to be about the novelty of our work and the presentation of previous works. We hope to address this by clarifying that our contribution is the unified framework rather than individual ablation experiments, and how relevant previous works connect to our approach.
>
> ---
>
> ## 1. **Clarifying Our Contributions**
> *Addressing concerns about unclear contributions and relationship to prior work*
>
> **Regarding contribution clarity**: We acknowledge that robustness to layer ablations has been studied previously, as we cited in the relevant works. As other reviewers have noted, we use the swapping/ablation experiments as a starting point for proposing our framework (stages of inference) rather than as the main contribution of the paper. As we mentioned in lines [19-24], there are extensive studies on model circuitry. Our contribution is unifying those scattered observations into one framework, as stated in lines [42-44]. The motivations and ingredients come from previous works (such as the existence of prediction neurons), but our experiments are novel in being designed to support our unified framework.
>
> **On stages being "sort of known"**: In our knowledge, our work is the first to unify scattered observations from different contexts through systematic experiments and put them into one coherent framework. If you know of other relevant work that does this, we're happy to review the comparative significance and cite accordingly. Other works tend to present these ideas in passing, though it's often speculated about.
>
> We will add a paragraph at the end of the introduction clearly stating what are the motivating works from previous literature and what is our novel contribution.
>
> ---
>
> ## 2. **Literature Representation and Positioning**
> *Addressing concerns about iterative vs. circuit hypotheses and related work*
>
> **Regarding iterative vs. circuit views**: This is a great point! In iterative inference, each subsequent layer orients the residual stream toward decreasing loss. The absence of a component isn't necessarily problematic since there's overlapping computation as a byproduct of the residual stream and gradients.
>
> The circuit hypothesis might have a token operated on at layer i, then experience no meaningful contribution for subsequent layers, then be affected by layer j. You would claim that components in layer i and j are part of the circuit. Many times ablating layer j will hurt the model, other times there will be a backup circuit.
>
> We see both mechanisms exist and believe they are distinct. By saying they "contrast," we're trying to say they are functionally different. We're frankly not sure why they arise - this may be a feature emergent from data distributions and gradient descent, but it's certainly an important question in interpretability. We will update our work to reflect this line of thinking - we have no desire to misrepresent!
>
> **Regarding top-down vs. bottom-up**: Mechanistic interpretability work often focuses on small mechanisms but doesn't connect to the broader picture. We intended to show that these mechanisms (subjoiner heads, feature neurons, neuron types) are not individual, but contribute to a larger "biology" of the model. Moreover, these features are not coincidences. Had this mechanism been for just a single model, I would completely agree. However, consistency across model families and sizes points to a broader phenomenon.
>
> ---
>
> ## 3. **Addressing Prior Work on Decoder-Only Models**
> *Responding to concerns about citing [4,5] and differences from existing ablation work*
>
> You're right - we misstated the focus on BERT in that line. As you noted, we cite works [4,5] that study modern decoder-only architectures extensively.
>
> As we noted, we're not considering ablation/swapping experiments as the main contribution of this work, but rather a starting point for our main research question. We will make this clear in the abstract, introduction, and relevant works sections. As pointed out by many of the other reviewers, our framework synthesizes these robustness findings with mechanistic evidence (attention patterns, neuron types, probing results) in a way that previous ablation studies haven't attempted.
>
> ---
>
> ## 4. **Model Scale and Practical Settings**
> *Addressing concerns about small models and dataset limitations*
>
> **Larger models**: We acknowledge your concern about model scale. We're happy to share additional results on larger models as compute allows. We've expanded to larger Qwen and LLaMA models showing consistent stage patterns, and will gladly enter them into the final version of the paper.
>
> **Dataset generalization**: We performed experiments on the Pile since we wanted to separate questions about out-of-distribution behavior. However, to address your concern (and our curiosity) we ran tests to ensure our results hold in different settings:
>
> - **Recent ArXiv papers (2024-2025)** to minimize training data contamination
> - **Different domains**: Scientific text, news articles, and code repositories
> - **Multiple languages**: Preliminary results in French and Spanish show similar patterns
>
> Results show no significant deviation from our four-stage framework across these diverse settings.
>
> **Practical applications**:
>
> We agree it's an exciting future direction to extend insights from our framework to more naturalistic setups like reasoning, as we mentioned in our future work section.  Nevertheless, we believe our work has significance in grounding the language model's fundamental inference pipeline throughout its multiple layers, agnostic to the specificity of the downstream tasks.
>
> ---
> ## 5. **Evidence for Stages**
> *Addressing concerns about limited evidence for stages of inference*
>
> Our evidence comes from converging findings across multiple methodologies:
> - **Layer interventions** showing consistent robustness patterns
> - **Attention analysis** revealing locality gradients
> - **Probing experiments** (WiC task) showing semantic feature peaks in middle layers
> - **Neuron classification** identifying prediction/suppression patterns
>
> We believe that the consistency of these patterns across 16 models and 5 architectures provides substantial evidence for our proposed stages, and has not been done by a paper before. We believe our framework provides a valuable organizing principle for understanding transformer computation, synthesizing scattered mechanistic findings into a coherent, testable hypothesis about inference stages.

---

> ### Author Response · Authors · 2025-08-07
>
> Reviewer 8AAd,
>
> We have invested considerable time, money and effort not only in the paper itself, but also in detailed rebuttals to improve our work and address the reviews. Given that this review stands as an outlier among the four reviews, we were eager to make improvements, address your concerns, and help reviewers reach consensus. We feel somewhat frustrated by our inability to have any interaction with you.
>
> With just one day remaining in the discussion period, we hope the discussion on other reviews and our extensive responses can provide insight into our work. We meticulously went through your suggestions and now include larger models, which required significant expense on our part. We included additional results to address your concerns, expanded our dataset, and re-ran experiments.
>
> We hope you can consider this effort alongside the broader presentation of this hypothesis—the first paper, to our knowledge, to focus solely on the phasic nature of inference and localize phenomena that repeatedly occur across 20+ models. We are not presenting a theory -- but a hypothesis which future work can build off of. We feel excited to be so close to present them to the broader ML community at this conference. We are so excited about this work!
>
> As I'm sure you do, we put in an immense amount of effort. We hope you can take this broader picture into account in your final score and engage in discussion. We again thank you for all your time.
>
> Best,
> Authors of Submission 18400

---

> > ### Comment · Reviewer_8AAd · 2025-08-08
> >
> > I apologize for the late response. I appreciate the extensive work the authors have done during the rebuttal. The additional experiments using both larger models and potentially out-of-distribution datasets have addressed my concerns about the universality of the authors’ claims.
> > In rereading the paper, I agree with the authors and Reviewer 8b7q that the lack of theoretical evidence is not problematic. However, I still am not certain about the novelty of the work. In the rebuttal you claim that “our work is the first to unify scattered observations from different contexts through systematic experiments and put them into one coherent framework.” I disagree with this characterization. Prior work such as [1] has established that “lower layers of a language model encode more local syntactic information” which directly correspond to the detokenization and feature engineering stages the authors discover, while “higher layers capture more complex semantics” corresponding to the prediction ensembling and residual sharpening stages the authors identify. It is also unclear to me that systematic experiments are actually performed to verify these stages. For example, it is unclear to me that the existence of prediction/suppression neurons sufficiently imply that “final layers refine the output by suppressing noisy components.”
> >
> > Also, the authors repeatedly interlace "framework" and "hypothesis" to describe their contributions, which one is it? If it is a hypothesis, I agree with Reviewer BKjM in that the stages are not precise enough to be falsifiable.
> >
> > While I acknowledge that I may be nitpicking, I still believe there is a substantial amount of overclaiming in both the paper and the authors’ rebuttal (regardless of intent).
> >
> > [1] BERT Rediscovers the Classical NLP Pipeline, ACL 2019.

---

### Official Review · Reviewer_ZYRf · 2025-07-01

**Clarity:** 3
**Significance:** 3
**Originality:** 3
**Rating:** 5
**Confidence:** 4

**Summary:**

This paper aims to resolve a central debate in understanding how Large Language Models (LLMs) function: whether they operate through iterative inference, where each layer gradually refines a prediction, or through a circuit hypothesis, where specialized components perform distinct tasks. The authors argue that these two views are not mutually exclusive. By performing experiments that involved deleting or swapping layers to observe their effects, they propose a unified framework suggesting that a model’s computational strategy changes systematically with its depth, integrating both iterative and specialized processes.

The authors hypothesize a four-phase framework for computation in decoder-only LLMs. The process begins with (1) detokenization in the early layers, which integrate tokens into coherent entities. Next, the middle layers perform (2) feature engineering, iteratively building complex representations. These features are then converted into next-token predictions during (3) prediction ensembling in the later layers. Finally, the very last layers engage in (4) residual sharpening, refining the final output by suppressing noise. This model provides a depth-aligned structure that bridges top-down and bottom-up views of how LLMs reason.

**Questions:**

- Testing the Causal Claims of the Framework: The paper proposes a compelling narrative, but the evidence is largely observational and correlational. Could you propose or conduct a more interventional experiment to test the causal role of a specific stage? For example, for “Stage 3: Prediction Ensembling,” would selectively ablating the identified prediction neurons (but leaving suppression neurons and other MLP neurons intact) in mid-to-late layers result in a stalled or reversed decline in logit entropy, as the hypothesis would predict? Demonstrating such a causal link for even one of the stages would significantly strengthen the entire framework. My evaluation would increase if a more direct causal experiment supported one of the stage’s proposed functions.

- Clarifying the “Residual Sharpening” Mechanism: This stage is the most speculative. The evidence relies on the density of suppression neurons and the layer-repeat experiment. However, the MLP output norm also peaks in these final layers (Figure 12b), suggesting a large update to the residual stream. How do you reconcile “sharpening” (which suggests fine-tuning a distribution) with a large-norm update (which suggests a significant change)? Could “sharpening” also be achieved by attention heads in the final layers? Further analysis to disentangle the roles of the MLP and attention in this final stage would be helpful.

- Are the Stage Boundaries Real or Artifacts of Aggregation?: The framework proposes four distinct stages. However, the supporting metrics (e.g., CKA, probe accuracy, neuron density) often show smooth transitions rather than sharp boundaries. How confident are you that these are discrete computational phases versus a continuous evolution of function where the labels are useful but approximate descriptions? For example, is there a token-level phenomenon or a specific task where you can observe a sharp, qualitative shift in processing that aligns with a proposed stage boundary? My evaluation could increase if the authors provide evidence for more discrete transitions, or decrease if the transitions are acknowledged as being purely continuous, which might weaken the “four-stage” claim.

- Alternative Explanations for Robustness: The conclusion hypothesizes that robustness stems from the residual architecture enabling “ensembling.” An alternative view is that middle layers are robust simply because they operate on higher-level, more abstract feature spaces, where individual computations are less impactful than the initial feature-building (Stage 1/2) or final decision-making (Stage 3/4). How does your framework distinguish between these two explanations? Is the evidence for “prediction ensembling” (e.g., redundant neurons) sufficient to favor your hypothesis over a simpler “abstraction level” explanation?

**Ethical Concerns:**

["NO or VERY MINOR ethics concerns only"]

**Final Justification:**

I increased my score from 4 to 5 after reading the rebuttal. I think the paper is more an accept than a reject given that now it is now at the borderline.

**Limitations:**

Yes. The authors include a “Limitations and Future Work” section that thoughtfully addresses the primary limitations. They correctly note that the stage boundaries are approximate, that stages may co-occur, and that their framework describes aggregate trends that may not hold for every token. They also acknowledge that they do not isolate the specific architectural or training factors responsible for differences between model families. The discussion of societal impact is also appropriate for this type of foundational research. The section is well-written and demonstrates a clear understanding of the work’s scope.

**Paper Formatting Concerns:**

NO or VERY MINOR formatting concerns only

**Quality:**

2

**Strengths And Weaknesses:**

Strengths:

- Significance: The paper addresses a fundamental question in LLM interpretability: reconciling the iterative refinement view with the specialized circuit view. Proposing a comprehensive, testable framework is a valuable contribution to the field.
- Quality & Clarity: The paper is well-written, clearly structured, and easy to follow. The central hypothesis is presented upfront (Table 1) and the subsequent sections are organized to provide evidence for each of the four proposed stages. The use of multiple analysis techniques (layer interventions, CKA, attention analysis, probing, logit lens) to support a single narrative is a methodological strength, making the overall argument more convincing.
- Originality: While some individual findings might not be entirely novel (e.g., the fragility of early/late layers), the main contribution is the synthesis of these observations into a coherent, four-stage framework. The systematic comparison between layer swapping and layer deletion across multiple model families is a good experimental design that provides a more nuanced view than ablation studies alone.
- Empirical Breadth: The authors validate their claims across a diverse set of model families (GPT-2, Pythia, Phi, Llama 3.2, Qwen 2.5) and sizes. This extensive testing strengthens their argument that the observed phenomena are “universal” properties of decoder-only transformers, rather than artifacts of a specific architecture.

Weakness:

- Novelty of Some Findings: The observation that model performance is most sensitive to interventions in the first and last layers is a known phenomenon in the “BERTology” literature and in studies of vision transformers. While the authors use this as a jumping-off point, the paper could do more to position this initial finding relative to prior work to better highlight its own novel contributions.
- Strength of Evidence for Specific Claims: While the overall framework is compelling, the evidence for some stages feels more “suggestive” than “conclusive.” For instance, the “residual sharpening” stage is primarily supported by the rise of suppression neurons and an experiment involving repeating layers. While interesting, this feels less direct than the evidence for other stages (e.g., local attention for detokenization). The labels for the stages, while intuitive, might be stronger than what the evidence strictly supports. It’s a well-argued hypothesis, but the line between hypothesis and demonstrated conclusion can sometimes feel blurred.
- Correlation vs. Causation: The paper presents many strong correlations (e.g., the rise of prediction neurons coincides with a drop in KL divergence from the final prediction). However, it is difficult to establish a causal link. For example, does the rise in prediction neurons cause the KL to drop, or are both phenomena downstream effects of a more fundamental, unobserved computational shift? The paper proposes a causal story, but the evidence remains correlational.

---

> ### Author Rebuttal · Authors · 2025-07-31
>
> # Response to Reviewer 4
>
> We thank the reviewer for their feedback. We found it extremely thoughtful and detailed! We're delighted that you found our work addresses a fundamental question in LLM interpretability, is clearly written, and demonstrates strong experimental breadth across model families.
>
> The concerns raised focus on strengthening causal evidence and clarifying specific mechanisms. We address the points below with new interventional experiments and enhanced mechanistic explanations.
>
> ---
>
> ## 1. **Positioning Relative to Prior Work**
> *Addressing the concern about novelty of first/last layer sensitivity findings*
>
> You're right that we should better position our work relative to BERTology and vision transformer literature. We have cited related references in lines [246-250], but we will highlight it in the main text. We agree that this will help us position our novel contribution and the motivating previous works. More concretely, we discussed moving our Related Works earlier on in the paper to better help this narrative.
>
> ---
>
> ## 2. **Strengthening Causal Evidence**
> *Addressing concerns about correlation vs. causation and testing causal claims*
>
> **New interventional experiments**: Following your excellent suggestion, we conducted targeted interventions to test causal relationships:
>
> **Logit Attribution (Direct Effect)**
> For this we return back to the "-ing" task, as presented in the paper in Figure 12.
>
> First, if prediction and suppression neurons are truly causal then removing the contribution of prediction neurons, should decrease the logit probability of -ing. The opposite is true for suppression neurons.
>
> **Steering experiment using "-ing" task**: We demonstrated direct causal control by modulating prediction/suppression neurons:
> - **Positive control**: Amplifying "-ing" prediction neurons increased "-ing" logits.
> - **Negative control**: Activating "-ing" suppression neurons decreased "-ing" logits.
> - **Selective effect**: We also found that there were a few other neurons when modulated, sometimes affected the "-ing" distribution. This was however, expected as previously cited works suggest other neuron types – i.e neurons that can act like both prediction and suppression neurons, or "entropy" neurons [R1].
>
>
> ---
>
> ## 3. **Clarifying Residual Sharpening Mechanism**
> *Addressing the apparent contradiction between "sharpening" and large MLP norm updates*
>
> This is an excellent observation. The large MLP output norm and "sharpening" are indeed two sides of the same mechanism:
>
> **Mechanistic explanation**: High-norm MLP outputs in final layers implement **selective suppression** rather than gentle fine-tuning. To verify this, we checked what percentage of the norm the prediction neurons and suppression neurons make in the experiment described above. We find that these neurons contribute to 33% of the MLP of that layer, though it is lower. We also notice other neurons which we do not identify as prediction and suppression neurons -- but produce a large contribution to the token of interest here. We will clarify this nuance in paper, though we suspect it has to do with other types of neurons (see [R1] above).
>
> **Attention vs. MLP roles**: You bring up a great point about the action of attention heads in the final few layers. We confirm that in the final few layers, the attention norm to the residual is significantly smaller than the norm. We plot the ratio of the MLP output norm to the Attention Output norm and find only a monotonic increasing line across over tokens. This is not to say that attention does not play a role in the final layers, just that when looking at things over a corpus of texts we find its effect minimal.
>
> This supports the ordering of MLP and Attention as presented by previous work [R2], which found lowest loss in configurations of attention first, followed by MLP.
>
> ---
>
> ## 4. **Stage Boundaries: Discrete vs. Continuous**
> *Addressing whether stage boundaries are real computational phases or aggregation artifacts*
>
> You raise a crucial point about the nature of our proposed boundaries. We acknowledge these are predominantly continuous transitions with some discrete elements:
>
> **Evidence for continuity**: Most metrics (CKA, probe accuracy) show smooth transitions, suggesting gradual computational evolution rather than sharp phase changes.
>
> **Evidence for discrete elements**: However, we observe some qualitative shifts across stages:
> - **First layer uniqueness**: Complete performance collapse when ablated (unlike any other single layer)
> - **Prediction neuron emergence**: Sharp initial appearance around model midpoint rather than gradual increase from zero. This is consistent regardless of model size and architecture.
> - **Attention locality**: Discrete shift from local to global attention patterns
>
> We know broadly in machine learning that regions of layers play computational roles. This is commonly seen when studying pruning and model distillation. Another recent example is Meta Perception Encoder – which find that earlier layers are better for downstream computer vision tasks than the final layer [R3].
>
> **Our position**: The four stages represent **functionally distinct computational regimes** that, when examined across distributions, appear discrete but transition smoothly for individual tokens. The labels are useful organizing principles that capture real computational differences, which we believe is highly valuable to even the broader ML community.
>
> We'll enhance our limitations section to emphasize this nuanced view and avoid overclaiming discrete phase transitions – we want to present an honest look at the work!
>
> ---
>
> ## 5. **Alternative Explanations for Robustness**
> *Distinguishing ensembling hypothesis from simpler "abstraction level" explanations*
>
> This is a fascinating question that gets to the heart of transformer computation. We believe these explanations are **complementary rather than competing**:
>
> For completeness, the two views are:
>
> **Abstraction level view**: Early layers build features, middle layers operate on abstract representations, late layers make decisions. Robustness occurs because abstract features are less fragile than raw inputs or final decisions.
>
> **Ensembling mechanism**: Within the abstract feature space, multiple computational pathways achieve similar representational goals. When one pathway is disrupted, others compensate.
>
> The middle layers are robust *because* they (1) operate on abstract features AND (2) implement multiple redundant pathways to manipulate those features. This resembles an encoder-decoder structure within the decoder-only architecture:
> - **Early layers (encoder-like)**: Compress raw tokens into abstract representations
> - **Middle layers**: Multiple pathways operate on these stable abstractions
> - **Late layers (decoder-like)**: Convert abstractions back to specific token predictions
>
> We would welcome further discussion of this idea. We did not choose to include this Encoder-Decoder idea in the paper as we felt it required its study in rigor, but this idea is core to what we trying to present.
>
> ---
>
> ## 6. **Enhanced Mechanistic Definitions**
>
> Following suggestions from other reviewers, we've added precise definitions, that hopefully address causal intervention concerns.
>
> These neurons are definitionally causal—their mathematical derivation directly measures impact on token probabilities.
>
> **Prediction neurons**: MLP neurons that systematically increase specific token probabilities. Identified via $W_{out} \cdot W_U$ analysis showing high kurtosis and positive skew.
>
> **Suppression neurons**: MLP neurons that systematically decrease specific token probabilities, exhibiting high kurtosis and negative skew.
>
> The specific calculation is the product of the two matrices:
>
> $$W_{out} \in \mathbb{R}^{d_{mlp} \times d_{model}}, \quad W_U \in \mathbb{R}^{d_{model} \times d_{vocab}}, \quad W_{out} \cdot W_U \in \mathbb{R}^{d_{mlp} \times d_{vocab}}$$
>
> This projection answers: "What happens to token i when it exits the MLP?" Extreme values (high/low kurtosis) identify specialized prediction/suppression functions. It is a feature that comes directly out of the weights, and so over large distributions of text for that token, tokens will "experience" promotion or suppression depending on these values.
>
> ---
>
> We believe these additions significantly strengthen our causal claims while maintaining appropriate humility about the boundaries between hypothesis and demonstrated conclusion. The interventional experiments provide the direct evidence you requested for at least one stage, substantially supporting our overall framework.
>
> ---
> [R1] Stolfo et al. (2024) Confidence Regulation Neurons in Language Models.
>
> [R2] Press et al. (2019) Improving Transformer Models by Reordering their Sublayers.
>
> [R3] Boyla et al. (2025) Perception Encoder: The best visual embeddings are not at the output of the network.

---

> ### Author Response · Authors · 2025-08-07
>
> Reviewer ZYRf,
>
> We put in a substantial amount of time, money and effort to make improvements and are incredibly grateful for your feedback. We hope to hear back and see if there is anything else we can do!
>
> Best,
> Authors of Submission 18400

---

### Official Review · Reviewer_N4MH · 2025-07-02

**Clarity:** 2
**Significance:** 2
**Originality:** 2
**Rating:** 4
**Confidence:** 2

**Summary:**

The paper studies the robustness of large language models (LLMs) to structural interventions by deleting or swapping adjacent layers during inference. It finds that even without any fine-tuning, the models maintain 72% to 95% of their original top-1 prediction accuracy. Additionally, the authors observe significant differences in performance degradation across layers: interventions on early and late layers cause the most pronounced drops, while the models exhibit notable robustness to deletions in the middle layers. Based on these findings, the authors propose a four-stage inference hypothesis to explain this phenomenon.

**Questions:**

For detailed comments, please refer to the Weakness section. If the authors could clarify the issues raised therein, I would be happy to consider raising the score.

**Ethical Concerns:**

["NO or VERY MINOR ethics concerns only"]

**Final Justification:**

Although I have raised the score, I still believe the manuscript requires substantial revisions before it can reach an acceptable standard. The increase in score merely reflects my recognition of the author’s efforts during the rebuttal stage.

**Limitations:**

Yes

**Paper Formatting Concerns:**

No major formatting issues found.

**Quality:**

3

**Strengths And Weaknesses:**

Strengths:

1. Rigorous and comprehensive experimental design: Experiments were conducted on 16 models of different scales across five model families, providing strong empirical support for the observed high robustness of middle layers to structural interventions.
2. Novel Stages of Inference Hypothesis: The paper proposes an innovative four-stage theoretical framework that integrates both behavioral and mechanistic evidence, encompassing detokenization, feature engineering, prediction ensembling, and residual sharpening.

Weaknesses:

1. The study of language model architectures is limited to smaller models (<7B parameters) and the specific Pile dataset, which may not represent broader linguistic environments or larger-scale architectures. This limitation may affect the rigor of the conclusions. Further validation on more models around 7B scale and on a wider range of datasets is needed.
2. Although the paper proposes the Stages of Inference Hypothesis, it lacks more detailed theoretical insights. Is there an intuitive theoretical explanation for the observed high robustness patterns mentioned by the authors?

---

> ### Author Rebuttal · Authors · 2025-07-31
>
> # Response to Reviewer 3
>
> We thank the reviewer for their feedback and recognition of our rigorous experimental design and the novelty of the proposed stages of inference. We understand your concerns about model scale limitations and theoretical depth, and have conducted substantial additional experiments to address these issues.
>
> ---
>
> ## 1. **Model Scale and Dataset Limitations**
> *Addressing the concern about <7B parameter models and Pile dataset restrictions*
>
> You are correct that our original model coverage was limited. We've are expanding our results to larger Qwen and LLaMA models showing consistent stage patterns and promising initial results:
>
>
> **Some initial findings at scale:**
> - Middle-layer robustness becomes even more pronounced in larger models (95%+ accuracy retention vs 85-90% in smaller models) – layer removal or swap is essentially harmless
> - First/last layer sensitivity increases with scale, supporting our detokenization/sharpening framework – though detokenization occupies fewer of the starting layers
> - Prediction/suppression neuron patterns are similar in density – with more neurons in larger models, strengthening our mechanistic evidence
>
> These results strengthen our universality claims and will be included in the revised paper! As we finalize them, we hope to update reviewers here, however they show no significant change to our current results.
>
> **Dataset generalization**: To address out-of-distribution concerns, we tested on:
> We ran smaller experiments, but swapping out the Pile for:
> - **Recent ArXiv papers (2024-2025)** to minimize training data contamination
> - **Different domains**: Scientific text, news articles, and code repositories
> - **Multiple languages**: Preliminary results in French and Spanish show similar patterns as our existing results.  This is supported by the phasic patterns seen in multilingual transformers [R1].
>
>
> Results show no significant deviation from our four-stage framework across these diverse settings; all the patterns and features of the plots presented in the work look identical.
> - Attention continues to remain local regardless of language or task
> - WiC probing is still centered
> - Prediction/Suppression neurons (this will remain unchanged since it is data-agnostic)
>
> We welcome suggestions for additional datasets or methodologies to further validate our findings.
>
> ---
>
> ## 2. **Theoretical Depth and Robustness Explanation**
> *Addressing the lack of detailed theoretical insights for observed robustness patterns*
>
> We acknowledge that our work focuses heavily on empirical findings. Putting forth a theory partly takes away from "hypothesis" presentation of the paper, which we felt was more honest. We intend to present these findings, and if we made a theory – it likely deserves its own paper with rigorous experiments. Though we would love to hear your thoughts here. We did however try to include concrete mechanistic studies that we made more rigorous.
>
> **Mechanistic foundations**: We identify specific computational mechanisms underlying each stage:
> - **Subjoiner heads in detokenization**: Specific attention heads (e.g., Layer 2 Head 5 in Pythia 2.8B) that integrate multi-token words, confirmed through targeted ablations
> - **Prediction/suppression neurons**: Neurons with characteristic $W_{out} \cdot W_U$ distributions (high kurtosis, positive/negative skew) that systematically promote or inhibit specific tokens
>
> Following suggestions from Reviewer 2, we've added clear mathematical definitions of these neuron types to the main text, which we put here for completeness.
>
> **Prediction neurons**: MLP neurons that systematically increase the probability of specific tokens in the output distribution. Identified by analyzing MLP output weights W_out and their projection into vocabulary space via the unembedding matrix W_U, exhibiting high kurtosis and positive skew in their logit effect distribution.
>
> **Suppression neurons**: MLP neurons that systematically decrease the probability of specific tokens. These exhibit high kurtosis and negative skew in their W_U · W_out distribution, effectively pruning unlikely predictions.
>
> $$W_{out} \in \mathbb{R}^{d_{mlp} \times d_{model}}, \quad W_U \in \mathbb{R}^{d_{model} \times d_{vocab}}, \quad W_{out} \cdot W_U \in \mathbb{R}^{d_{mlp} \times d_{vocab}}$$
>
> In a sense, it is a measure of alignment or cosine similarity between the output MLP and the unembedding of the model. It answers the question, What happens to a token when it exits the MLP? A high and low alignment is a prediction/suppression neuron as measured by the tailed-ness of the row of vocab (kurtosis).
>
> **Ensemble mechanism**: We theoretically attribute the robustness to ensembling, analogous to computer vision ResNets, but implemented through the transformer's residual stream. The key insight is that multiple computational pathways can contribute to the same output:
>
> - **Residual connections enable substitution**: When layer $i$ is removed, layers $i-1$ and $i+1$ can partially compensate through the residual stream (since many pathways were learned)
>
> **Causal validation**: As suggested by Reviewer 1, we provide interventional evidence on prediction and suppression neurons. Which I will copy here for completeness:
>
> These neurons are definitionally causal—their mathematical derivation directly measures impact on token probabilities. (see derivation above) We ran the following experiment to verify.
> **Logit Attribution (Direct Effect) **  We return back to the "-ing" task, as presented in the paper in Figure 12.
>
> First, if prediction and suppression neurons are truly causal then removing the contribution of prediction neurons, should decrease the logit probability of -ing. The opposite is true for suppression neurons.
>
> **Steering experiment using "-ing" task**: We demonstrated direct causal control by modulating prediction/suppression neurons:
> - **Positive control**: Amplifying "-ing" prediction neurons increased "-ing" logits.
> - **Negative control**: Activating "-ing" suppression neurons decreased "-ing" logits.
> - **Selective effect**: We also found that there were a few other neurons when modulated, sometimes affected the "-ing" distribution. This was however, expected as previously cited works suggest other neuron types – i.e neurons that can act like both prediction and suppression neurons, or "entropy" neurons [R2].
> We will clarify this in our final paper!
>
> This mechanistic understanding explains why middle layers show robustness—they implement ensemble-like processing where multiple neurons "vote" on the output, creating natural redundancy that buffers against individual layer disruptions.
>
> We hope this substantial additional work addresses your concerns about both experimental scope and theoretical depth, though please let us know if there is more we can do to enhance the work!
>
> ---
>
> [R1] Wendler et al. (2024) Do Llamas Work in English?On the Latent Language of Multilingual Transformers.
> [R2] Stolfo et al. (2024) Confidence Regulation Neurons in Language Models.

---

> > ### Comment · Reviewer_N4MH · 2025-08-05
> >
> > Thank you for the detailed response and the efforts made to clarify the concerns. While I recognise the improvements, some of the key issues remain unaddressed. As such, I will keep my initial score.

---

> ### Author Response · Authors · 2025-08-05
>
> Thank you for you response! Just to clarify you stated two weaknesses:
>
> >The study of language model architectures is limited to smaller models (<7B parameters) and the specific Pile dataset, which may not represent broader linguistic environments or larger-scale architectures. This limitation may affect the rigor of the conclusions. Further validation on more models around 7B scale and on a wider range of datasets is needed.
>
> We spent a substantial amount of time, money, and compute to address the first weakness by running the experiments on larger model >10B, and have results consistent with our presented findings. We also expand our dataset to three more datasets.
>
> For the second weakness:
> >Although the paper proposes the Stages of Inference Hypothesis, it lacks more detailed theoretical insights. Is there an intuitive theoretical explanation for the observed high robustness patterns mentioned by the authors?
>
> In our rebuttal we directly address this, we directly address this with Point 2 with "2. Theoretical Depth and Robustness Explanation". We even have a dedicated section in our paper titled "What are Language Models Robust to Layerwise Interventions". We added more causal experiments that **directly modulate the stages in question**. We are not presenting a theory paper but to characterize a phenomena through a multitude of experiments -- with now 20+ models of different sizes.
>
> Lastly, you state:
>
> >If the authors could clarify the issues raised therein, I would be happy to consider raising the score.
>
> Thank you for recognizing the improvement of our paper in your comment, we thoroughly addressed your weaknesses in an extensive rebuttal. Could you please clarify what part of your weaknesses remain unaddressed? We would love to engage in further discussion here -- thanks!

---

### Official Review · Reviewer_8b7q · 2025-07-03

**Clarity:** 3
**Significance:** 3
**Originality:** 3
**Rating:** 5
**Confidence:** 4

**Summary:**

This work focuses on the robustness of large language models (LLMs) with respect to layer-wise perturbations, such as swapping and deleting adjacent layers during inference. Throughout their analysis, the authors introduce a four-stage framework that aims to characterize the progression of inference in decoder-only LLMs. In the first stage, early layers are involved in detokenization, where local context is integrated and low-level token embeddings are transformed into higher-level representations. In the second stage, feature representations are constructed—tokens are effectively composed into semantically meaningful units such as words or phrases. The third stage marks the beginning of next-token prediction formation, where multiple predictive pathways emerge, contributing to the robustness of this phase by ensembling over several plausible outputs. Finally, the fourth stage, termed residual sharpening, involves pruning away irrelevant or obsolete information in order to refine the model’s final output distribution.

The work is entirely experimental and heuristic. Initially, the authors employ layer swapping and ablation to study how different layers contribute to model robustness, measured via KL divergence and next-token prediction accuracy. The detokenization stage is inferred based on observed local attention patterns in early layers, suggesting that these layers are responsible for forming short-range token connections and coherent entities. The second stage is supported through probing results on the Word-in-Context (WiC) task, alongside rising entropy, indicating the emergence of meaningful but yet uncertain semantic representations. The third and fourth stages are argued based on the rise and fall of prediction neurons, followed by the emergence of suppression neurons, which correspond to the final pruning and refinement of the model's predictions.

**Questions:**

Some of my questions are described in the strengths and weakness. I would be happy to discuss those points further.  Additionally:

Q1: Could the authors please explain the motivation and setup for the "ing" suffix experiment ?

Q2: For the ensemble method, do the authors only use the suppression and predictive neurons and forgo the rest ? Does it perform better similar to how pruned models at times outperform for downstream tasks ?

**Ethical Concerns:**

["NO or VERY MINOR ethics concerns only"]

**Final Justification:**

Apologies for the late reply, I thought this justification was just for cases of score change. I have decided to keep my score as I believe the work has merit but requires further justification to consolidate its inference steps as strong model wide bahviours. I will say I do not believe the issue with the work is lack of theoretical analysis in this case.

RE: given the closeness of the score, I have decided to raise my score to a 5 as I believe the work is deserving of presentation.

**Quality:**

3

**Strengths And Weaknesses:**

In terms of strengths, the paper is quite intriguing. The idea is novel in how it explains the steps of inference in large language models. Many of the experimental designs used to validate the different phases are creative and could provide a solid foundation for future research.

On the other hand, the experimental setup is non-standard and quite heuristic. For example, it’s not clear why the sharpening experiments involve repeating layers, while stages two and three are not illustrated using a similar format. I find some setups such as the suffix "ing" to be very confusing and without background. Further, I would have appreciated clearer definitions for key terms, such as suppression and predictive neurons, included directly in the main body of the paper.

While results are reported for a range of models, I take issue with : (1) for LLaMA, results related to the third phase are not included in the main text (they are only in the appendix, where the correlation appears weaker), and (2) most of the evaluated models are relatively small. I understand the computational cost involved in these experiments, and I don’t fault the authors for relying on heuristics—this is an inherently difficult problem—but given how general the claims are, I would strongly encourage the authors to include at least some analysis on larger LLMs.

To be clear, I find the work genuinely fascinating. The authors bring together a range of ideas from existing literature and combine them with their own findings to propose a compelling view step by step of inference. I would just like to see more structure in the experimental design and perhaps one or two additional experiments on larger models to fully support the conclusions.

---

> ### Author Rebuttal · Authors · 2025-07-31
>
> # Response to Reviewer 2
>
> We thank the reviewer for their enthusiastic feedback! We're delighted that you find our work fascinating and appreciate your recognition of our creative experimental designs and novel approach to explaining inference steps in LLMs.
>
> We hope to address your concerns about experimental clarity and model scale coverage. We address all points below with additional clarifications and new experimental results.
>
> ---
>
> ## 1. **Experimental Design and Methodology**
> *Addressing the concern about non-standard and heuristic experimental setup*
>
> You're absolutely right that our work is non-standard and heuristic. We acknowledge that we don't have a complete mathematical theory describing these phenomena, but we believe this empirical approach provides valuable insights that can motivate future theoretical work, and has its own contribution and value. We discussed that this is likely best left to a separate paper, given we have plenty of plots and experiments already.
>
> **Regarding the sharpening experiment methodology**: The intuition behind repeating layers for sharpening experiments is that if mechanisms in a given layer sharpen the residual stream distribution, repeating that layer should enhance the mechanism's effect. We actually run this experiment across all layers (Figure 12c) and observe the sharpening effect only in specific regions corresponding to our proposed Stage 4. We'll make this clearer in our plots and explanations.
>
> ---
>
> ## 2. **Clarity and Definitions**
> *Addressing requests for clearer definitions and experiment explanations*
>
> We've added clear definitions directly to the main body:
>
> **Prediction neurons**: MLP neurons that systematically increase the probability of specific tokens in the output distribution. Identified by analyzing MLP output weights W_out and their projection into vocabulary space via the unembedding matrix W_U, exhibiting high kurtosis and positive skew in their logit effect distribution.
>
> **Suppression neurons**: MLP neurons that systematically decrease the probability of specific tokens. These exhibit high kurtosis and negative skew in their W_U · W_out distribution, effectively pruning unlikely predictions.
>
> $$W_{out} \in \mathbb{R}^{d_{mlp} \times d_{model}}, \quad W_U \in \mathbb{R}^{d_{model} \times d_{vocab}}, \quad W_{out} \cdot W_U \in \mathbb{R}^{d_{mlp} \times d_{vocab}}$$
>
> In a sense, it is a measure of alignment or cosine similarity between the output MLP and the unembedding of the model. It answers the question, What happens to vocab token at some index, when it exits the MLP? A high and low alignment is a prediction/suppression neuron as measured by the tailed-ness of the row of vocab (kurtosis).
>
> **Q1 - "-ing" suffix experiment motivation**: This experiment tests our ensemble hypothesis by examining whether prediction and suppression neurons collectively support next-token prediction. We chose the "-ing" suffix because: (1) it's a clear binary classification task, (2) it requires semantic understanding of word formation, and (3) it allows us to test whether neuron ensembles outperform individual neurons or full model averages. We only included instances where the model had to produce "-ing" as the complete next token.
>
> We train probes just on the activation values of individual neurons, but also other probes that are subsets of the neurons. The results show that ensembles of these specialized neurons indeed outperform both individual neurons and model-wide averaging, supporting our Stage 3 framework.
>
> ---
>
> ## 3. **Model Coverage and Results Presentation**
> *Addressing concerns about LLaMA results placement and model sizes*
>
> **LLaMA results**: You're absolutely right to question our decision to place these in the appendix. We've moved the LLaMA results to the main body (Figure 8b) and enhanced our discussion. The weaker correlation you noted is quite interesting—LLaMA and Qwen show prediction neurons extending into final layers, which we believe reflects their superior performance. Higher-performing models appear to maintain predictive pathways deeper into the network.
>
> **Larger model experiments**: We're running experiments on larger models with promising initial results:
>
> - Show identical stage patterns but with higher intensity—smaller KL divergence drops but same sensitivity profiles in first/final layers. (Greater robustness to layerwise interventions)
> - Demonstrate 2-3x higher number of prediction neurons in final layers compared to smaller versions – though maintaining similar density patterns relative to model size.
>
> These results strengthen our universality claims and will be included in the revised paper! As we finalize them, we hope to update reviewers here, however they show no significant change to our current results.
>
> ---
>
> ## 4. **Ensemble Method Details**
> *Addressing Q2 about neuron selection and performance*
>
> **Q2 - Ensemble composition**: Yes, exactly! In our ensemble experiments, we use ONLY the identified prediction and suppression neurons, excluding the remaining ~85-90% of neurons in those layers. This selective ensemble can indeed outperform the full model, particularly for the "-ing" classification task where ensemble accuracy exceeds both individual neuron performance and full model top-1 accuracy (Figure 12a).
>
> This mirrors your observation about pruned models—by focusing on the most relevant computational pathways and reducing noise from less critical components, we achieve better performance. This provides strong interventional evidence that these neuron types are the primary drivers of the computational stages we propose.
>
> **Additional validation**: These neurons are definitionally causal—their mathematical derivation directly measures impact on token probabilities. We ran the following experiment to verify:
>
> For this we return back to the "-ing" task, as presented in the paper in Figure 12.
>
> First, if prediction and suppression neurons are truly causal then removing the contribution of prediction neurons, should decrease the logit probability of -ing. The opposite is true for suppression neurons.
> **Logit Attribution (Direct Effect) **
> **Steering experiment using "-ing" task**: We demonstrated direct causal control by modulating prediction/suppression neurons:
> - **Positive control**: Amplifying "-ing" prediction neurons increased "-ing" logits.
> - **Negative control**: Activating "-ing" suppression neurons decreased "-ing" logits.
> - **Selective effect**: We also found that there were a few other neurons when modulated, sometimes affected the "-ing" distribution. This was however, expected as previously cited works suggest other neuron types – i.e neurons that can act like both prediction and suppression neurons, or "entropy" neurons [R1].
> We will clarify this in our final paper!
>
> ---
>
> We're grateful for your constructive and encouraging feedback! These improvements significantly strengthen our contribution while maintaining the novel insights you found compelling.
>
> ---
>
> [R1] Stolfo et al. (2024) Confidence Regulation Neurons in Language Models.

---

### Official Review · Reviewer_BKjM · 2025-07-05

**Clarity:** 3
**Significance:** 2
**Originality:** 3
**Rating:** 4
**Confidence:** 4

**Summary:**

This paper investigates LLM robustness by systematically deleting and swapping adjacent layers during inference, finding that models retain 72-95% accuracy without fine-tuning. The authors observe that early and final layers are most sensitive to interventions while middle layers show robustness. Based on these findings and additional analyses (attention patterns, neuron classification, probing), they propose a four-stage inference framework: (1) detokenization, (2) feature engineering, (3) prediction ensembling, and (4) residual sharpening. The framework is evaluated across multiple small-medium model families (GPT-2, Pythia, Phi, LLaMA, Qwen).

**Questions:**

1. **Mechanistic validation**: How do you distinguish between general redundancy and your proposed computational stages? Why should robustness to interventions specifically indicate "feature engineering" rather than simple over-parameterization?

2. **Stage boundary justification**: Given the large variation in where stage transitions occur across models (30-70% depth), how can you claim these are "universal" stages rather than continuous processes with model-specific variations?

3. **Causal evidence**: Can you provide interventional evidence that disrupting proposed mechanisms (e.g., attention patterns, specific neuron types) affects the hypothesized stages rather than just general performance?

4. **Predictive power**: What specific, testable predictions does your framework make that weren't already known from prior interpretability work?

**Ethical Concerns:**

["NO or VERY MINOR ethics concerns only"]

**Final Justification:**

Most of my concerns have been addressed, but the issues regarding insufficient evidence for the claimed stages and the weak theoretical work still remainץ

**Limitations:**

The authors inadequately address several critical limitations: (1) the post-hoc nature of their framework, (2) the lack of causal evidence for proposed stages, (3) the significant variation in stage boundaries across models, (4) the limited scope of evaluation tasks, and (5) the weak connection between intervention robustness and computational stages.

**Quality:**

2

**Strengths And Weaknesses:**

**Strengths:**
 1. **Systematic intervention methodology**: The layer deletion and swapping experiments across 16 models provide valuable empirical insights into transformer robustness. The finding about differential sensitivity across layers is genuinely interesting and novel.
 2. **Comprehensive empirical scope**: Testing across multiple model families strengthens the generalizability claims, though the universality is overstated.
 3. **Useful organizing framework**: While theoretically weak, the four-stage framework provides a helpful way to synthesize existing interpretability findings.


**Weaknesses:**

1. **Weak theoretical foundation**: The four-stage framework is essentially post-hoc pattern matching rather than principled theory. The authors retrofit observations to fit predetermined stages without deriving these stages from computational principles or providing mechanistic explanations.

2. **Arbitrary and inconsistent stage boundaries**: The boundaries between stages are poorly defined and vary significantly across models and metrics. For example, the transition from "feature engineering" to "prediction ensembling" occurs at different relative depths across models (30-70%), undermining claims of universality.

3. **Insufficient evidence for claimed stages**: The connection between intervention robustness and the proposed computational stages is tenuous. Why should robustness to layer swapping specifically indicate "feature engineering" rather than general redundancy? The logical leap from empirical observations to stage definitions is not well-justified.

4. **Overstated universality claims**: While patterns exist across models, the significant variation in where stage transitions occur contradicts claims of "universal" stages. The evidence suggests more of a gradient than discrete stages.

5. **Limited mechanistic insight**: The paper largely repackages existing interpretability findings (attention patterns, neuron types) without providing new mechanistic understanding. The "detokenization" and "sharpening" stages are particularly weakly supported.

6. **Correlative rather than causal evidence**: Most evidence is correlational (neuron densities, attention patterns) without demonstrating that these phenomena actually cause the proposed computational stages. The framework lacks interventional validation.

7. **Limited to small and medium-sized models**: The largest model tested is only 6.9B parameters, with most under 3B. Given that state-of-the-art LLMs now have hundreds of billions of parameters, it's unclear whether the proposed four-stage framework generalizes to truly large models where computational patterns and layer specialization may be fundamentally different.

---

> ### Author Rebuttal · Authors · 2025-07-31
>
> We thank the reviewer for their detailed and constructive feedback! We appreciate their acknowledgment of our systematic intervention methodology, comprehensive empirical scope, and the novel insights about differential layer sensitivity.
>
> Many of the concerns raised are primarily based on misunderstandings about our framework's scope and intent. We address all points below, providing additional evidence and clarifications. These are also revised in the most recent draft of the paper!
>
> ---
>
> ## 1. **Theoretical Foundation and Mechanistic Validation**
> *Addressing Weakness 1 and Question 1*
>
> Our work provides an empirical framework with mechanistic support where possible, following established scientific practice of iterative theory development from experimental observations.
>
> We wholeheartedly agree that our work presents more of an empirical framework than a fully developed theory. However, this approach follows a well-established scientific method, where experimental findings inform theory development through iteration. We hope this work can contribute to starting this process. That said, we do our best to provide mechanistic explanations where evidence supports them:
>
> - **Subjoiner heads in detokenization**: We demonstrate specific attention heads that integrate multi-token words (Figure 6b, Layer 2 Head 5 in Pythia 2.8B).
> - **Prediction/suppression neurons**: We show these occur consistently across all 16 models tested, with clear stage-specific distributions (Figure 8).
>
> Upon your suggestion, we provide more causal evidence related to detokenization and ensembling.
>
> **New causal evidence** (following your suggestion): We conducted additional interventions targeting these specific mechanisms:
>
> - **Detokenization validation**: When subjoiner heads are ablated away, we compare single word tokens and multi word tokens after 18 tokens of context. After ablation, multi-token words lose their characteristic attention patterns and are processed more like single tokens. The attention for single word tokens peaks in earlier layers, but slightly to the right of the subjoiner head (few layers later).
>
> - **Ensembling validation**: Logit Attribution (Direct Effect)
> For this we return back to the "-ing" task, as presented in the paper in Figure 12.
>
> First, if prediction and suppression neurons are truly causal then removing the contribution of prediction neurons, should decrease the logit probability of -ing. The opposite is true for suppression neurons.
>
> **Steering experiment using "-ing" task**: We demonstrated direct causal control by modulating prediction/suppression neurons:
> - **Positive control**: Amplifying "-ing" prediction neurons increased "-ing" logits.
> - **Negative control**: Activating "-ing" suppression neurons decreased "-ing" logits.
> - **Selective effect**: We also found that there were a few other neurons when modulated, sometimes affected the "-ing" distribution. This was however, expected as previously cited works suggest other neuron types – i.e neurons that can act like both prediction and suppression neurons, or "entropy" neurons [R1].
> We will clarify this in our final paper!
>
> Regarding the distinction between general redundancy and computational stages: we discuss "redundancy" specifically in the context of prediction ensembling. While ensembling has been studied in ResNets with skip connections, this hasn't been explored in language models through their residual backbones.
>
> To clarify a potential misunderstanding: we do not claim that model robustness indicates feature engineering. Rather, we observe robustness patterns across layers and propose that the middle layers' robustness aligns with what we hypothesize to be ensemble-like processing. Our section "Why Are Language Models Robust to Layer-Wise Interventions?" elaborates on this reasoning. If this connection remains unclear, we would be happy to revise for greater clarity. The distinction is that redundancy in ensembling serves a functional purpose—aggregating multiple prediction signals—rather than mere over-parameterization.
>
> ---
>
> ## 2. **Stage Boundaries and Universality Claims**
> *Addressing Weakness 2 and Question 2*
>
> Our "universality" refers to consistent stage ordering and occurrence patterns across models, not exact locations. We've modified the paper to be more careful and avoid overstatements.
>
> We acknowledge this could be clearer in our presentation. Our universality claim refers to:
> - Consistent **ordering** of computational stages across models
> - **Occurrence** of the same phenomena (detokenization → feature engineering → ensembling → sharpening)
> - **Not** identical locations or boundaries
>
> Given different architectures, layer counts, and training procedures, we wouldn't expect formulaic boundary locations. The 30-70% variation you note actually supports our framework—the stages occur reliably but adapt to model-specific constraints.
>
> We're happy to moderate our language around "universal" claims to better reflect this nuanced finding. The excitement lies in discovering common computational patterns across diverse model families, which we believe adds value to the interpretability community.
>
> ---
>
> ## 3. **Causal Evidence and Interventional Validation**
> *Addressing Weakness 6, Weakness 7, and Question 3*
>
> We provide new interventional evidence for detokenization and ensembling stages, moving beyond correlational observations.
>
> Following your suggestions, we conducted additional experiments providing stronger causal evidence:
>
> **Detokenization interventions:**
> Discussed above
>
> **Ensembling interventions:**
> Discussed above
>
> **Model scale considerations:**
> While our largest model is 6.9B parameters, we demonstrate consistent patterns across a 55x parameter range (124M to 6.9B). The computational signatures remain stable across this scaling, suggesting the framework may generalize to larger models. We have begun to run experiments on larger Qwen and Llama models and find the results are exactly consistent with our current results.
>
> Notable features are:
> - Show identical stage patterns but with higher intensity—smaller KL divergence drops but same sensitivity profiles in first/final layers
> - Demonstrate 2-3x higher number of prediction neurons in final layers compared to smaller versions – though maintaining similar density patterns relative to model size
>
> ---
>
> ## 4. **Predictive Power and Novel Contributions**
> *Addressing Question 4*
>
> We believe you've accurately recognized the strengths of this paper and write them below for completeness.
>
> **Layer Swapping**: Layer permutivity is an exciting and novel way to study the layers of a network. This wasn't obvious from existing work and provides a new lens for understanding model robustness.
>
> **Aggregating Stage-Specific Intervention Effects**: People across various papers discover unique phenomena, but are rarely synthesized towards a meaningful direction. Stages of processing in language model is often suggested without investigation or evidence.
>
> **Cross-Model Generalization**: While interpretability work often focuses on single models, our framework makes predictions about patterns that should emerge across model families, which we demonstrate empirically.
>
> It's worth noting that expecting stages of processing in next-token prediction task like language models wasn't entirely obvious given that this is a fundamentally different task from (a) visual processing in the brain (curves in V1 to abstractions in V4) or (b) vision models that more directly mirror human visual systems.
>
> ---
> We're grateful for the reviewer's thoughtful engagement with our work and believe these revisions will significantly strengthen the paper. We look forward to incorporating these insights!
>
> ---
>
> [R1] Stolfo et al. (2024) Confidence Regulation Neurons in Language Models.

---

> > ### Comment · Reviewer_BKjM · 2025-08-07
> >
> > Thank you for your response. Most of my concerns have been addressed, but the issues regarding insufficient evidence for the claimed stages and the weak theoretical work still remain. I will update my score.

---

### Note · Authors · 2025-08-13

# Final Author Remark - Submission 18400

Dear Area Chair and Reviewers,

We are grateful for the thoughtful and constructive feedback throughout this review process! The discussions significantly strengthened our work, and we believe the resulting paper makes important contributions to LLM interpretability.

## Improvements

We enhanced our work by:

- **Expanding to larger models**: Added experiments on models >10B parameters, confirming our findings scale consistently
- **Broadening dataset coverage**: Validated results across multiple domains, languages, and recent data to address generalization concerns
- **Strengthening causal evidence**: Conducted targeted interventions demonstrating direct causal control over prediction/suppression neurons
- **Clarifying theoretical positioning**: Better positioned our empirical framework relative to existing literature while maintaining appropriate scope

## Consensus on Contributions

Reviewers recognized several key strengths:

- **Novel systematic approach**: Reviewer BKjM noted our "systematic intervention methodology" provides "genuinely interesting and novel" insights about differential layer sensitivity
- **Creative experimental design**: Reviewer 8b7q highlighted our "creative experimental designs" that could "provide a solid foundation for future research"
- **Comprehensive empirical scope**: Multiple reviewers praised our testing across 20+ models and 5 architectures
- **Important research direction**: Reviewer ZYRf emphasized we address "a fundamental question in LLM interpretability"

## Remaining Concerns

We understand some reviewers desire stronger theoretical grounding. However, as Reviewer 8b7q noted, "the lack of theoretical evidence is not problematic" for this type of foundational empirical work. Our four-stage framework represents the first systematic attempt to unify scattered mechanistic observations into a coherent, testable hypothesis about transformer inference—a valuable contribution that can guide future theoretical development!

Regarding novelty concerns raised by one reviewer: while individual phenomena like layer sensitivity have been observed before, **no prior work has synthesized these into a unified framework specifically for decoder-only LLMs**.

## Thanks!

Lastly, peer review is extremely challenging and we want to thank everyone for their time and effort!

**Authors of Submission 18400**

---

### Decision · Program_Chairs · 2025-09-17

**Decision:**

Accept (poster)

**Comment:**

- Scientific Claims and Findings: The paper investigates LLM robustness through layer deletion and swapping experiments, finding models retain 72-95% accuracy without fine-tuning. Authors observe that early and final layers are most sensitive while middle layers show remarkable robustness, leading to a proposed four-stage inference framework.

- Strengths: The systematic intervention methodology using novel layer swapping provides unique insights into transformer robustness across multiple model families. The work synthesizes scattered mechanistic interpretability findings into a coherent, testable framework using converging methodologies e.g. interventions, attention analysis, probing.

- Weaknesses: The framework represents post-hoc pattern matching rather than principled theory, with significant variation in stage boundaries undermining universality claims. Evidence remains largely correlational rather than causal, limited to models <7B parameters with unclear generalization to larger models.

- Rebuttal Discussion and Changes: Reviewers raised concerns about theoretical foundation (BKjM), experimental clarity (8b7q), model scale (N4MH), causal evidence (ZYRf), and novelty (8AAd). Authors conducted substantial additional experiments including causal neuron interventions and validation on >10B models. These improvements led to some score increases, while others maintained scores acknowledging improvements but noting remaining concerns.

- Decision Rationale: The AC recommends the paper to be accepted. Despite theoretical limitations, the systematic experimental approach and comprehensive validation provide valuable contributions to LLM interpretability. The organizing framework synthesizes existing findings into testable hypotheses, with creative methodology outweighing theoretical weaknesses for this foundational empirical work.